

# Snowmelt response to simulated warming across a large elevation
# gradient, southern Sierra Nevada, California
Keith N. Musselman[1], Noah P. Molotch[2], and Steven A. Margulis[3]
[1]National Center for Atmospheric Research, Boulder, Colorado, USA
[2]Department of Geography, Institute of Arctic and Alpine Research, University of Colorado,
Boulder, USA & Jet Propulsion Laboratory, California Institute of Technology, Pasadena, USA
[3]Department of Civil and Environmental Engineering, University of California, Los Angeles,
California, USA
*Correspondence to*: Keith N. Musselman (kmussel@ucar.edu)



## 10    Abstract

In a warmer climate, the fraction of annual meltwater produced at high melt rates is projected to
decline due to a contraction of the melt season to an earlier period of lower energy. How
snowmelt rates, including extreme events relevant to flood risk, may respond to a range of
warming over a mountain front remains poorly known. We present a model sensitivity study of
snowmelt response to warming across a 3600 m elevation gradient in the southern Sierra
Nevada, USA. A snow model was run for three distinct years and verified against extensive
ground observations. To simulate the impact of climate warming on meltwater production,
measured meteorological conditions were modified by +1°C to +6°C. The total annual snow
water volume exhibited linear reductions (-10% $°C^{-1}$) consistent with previous studies. However,
the sensitivity of snowmelt rates to successive degrees of warming varied nonlinearly with
elevation. Middle elevations and years with more snowfall were prone to the largest reductions
in snowmelt rates, with lesser changes simulated at higher elevations. Importantly, simulated
warming causes extreme daily snowmelt (99[th] percentiles) to increase in spatial extent and
intensity and shift from spring to winter. The results offer critical insight into the sensitivity of
mountain snow water resources and how the rate and timing of water availability may change in
a warmer climate. The identification of future climate conditions that may increase extreme melt
events is needed to address the climate resilience of regional flood control systems.



## 1. Introduction

Seasonal snow accumulation and melt in mountainous areas are critical components of the
regional hydrologic cycle with important controls on climate, ecosystem function, flood risk, and
water resources [*Bales et al.*, 2006; *Barnett et al.*, 2005]. Warmer temperatures are expected to
reduce snowpack volume and persistence [*Gleick*, 1987; *Knowles and Cayan*, 2004; *Mote et al.*,
2005] by shifting precipitation from snowfall to rain [*Knowles et al.*, 2006] and causing earlier
snowmelt [*Stewart et al.*, 2004]. Studies of historical observations in the western U.S. have
identified recent declines in spring snowpack [*Mote et al.*, 2005], diminished snowmelt runoff
volumes [*Dettinger and Cayan*, 1995; *McCabe and Clark*, 2005] and earlier spring runoff
[*Stewart et al.*, 2004]. Most of these studies have attributed the observed trends to anomalously
warm spring and summer temperatures of recent decades. *Fyfe et al.* [2017] report that the recent
snowpack declines are not replicable with climate model simulations forced by natural changes
(i.e., internal variability) alone, but are resolved when both natural and anthropogenic changes
are considered.

Continued warming is expected. General Circulation Models (GCMs) project increases in

global average temperatures ranging from 0.7°C ± 0.4°C to 6.5°C ± 2.0°C for the lowest and
highest greenhouse gas emission scenarios, respectively, for the end of the next century [*Stocker*
*et al.*, 2013]. The effects of a warmer climate on the snow-dominated hydrology of the Sierra
Nevada are generally recognized to include higher winter storm runoff and flood risk, and
reduced summer low-flows [*Dettinger*, 2011; *Dettinger et al.*, 2004; *Godsey et al.*, 2013;
*Knowles and Cayan*, 2002; *Lettenmaier and Gan*, 1990]. It is not well understood how present-
day snowmelt rates may respond to the range of projected warmer temperature scenarios and,
particularly, how those changes will impact water availability over large elevation gradients.



Elevation is a dominant explanatory variable of mountain snow-cover persistence
[*Girotto et al.*, 2014b], ranking in importance above solar radiation and terrain aspect for many
basins in the western U.S. [*Molotch and Meromy*, 2014]. Snowpack response to warmer
temperatures exhibits strong nonlinear elevation dependencies [*Brown and Mote*, 2009; *Knowles*
*and Cayan*, 2004]. For example, slight warming can cause drastic hydrologic response at lower
elevations as rain becomes the predominant hydrologic input and snow-cover becomes
seasonally intermittent or negligible [*Hunsaker et al.*, 2012; *Marty et al.*, 2017; *Nolin and*
*Daly*, 2006]. At higher and cooler elevations, snowmelt may remain a substantial component of
the annual hydrologic input in a warmer climate, but the timing and rate of melt is altered. Rapid
and prolonged spring snowmelt is unique to these mountain environments [*Trujillo and Molotch*,
2014]. This efficient runoff generation mechanism [*Barnhart et al.*, 2016] produces water
resources of vast economic importance [*Sturm et al.*, 2017]. Improved understanding of regional
elevation-dependent snowmelt response to warming is a key step toward better predicting and
interpreting model estimates of basin-wide runoff.
In a warmer climate, the fraction of meltwater produced at high melt rates is projected to
decrease due to a contraction of the historical melt season to a period of lower available energy
[*Musselman et al.*, 2017]. Because streamflow is a nonlinear response to hydrologic input, slight
reductions in snowmelt rates may disproportionately reduce runoff. Despite recent advances in
process understanding, the sensitivity of snowmelt rates to a range of potential warming over a
foothills-to-headwaters elevation profile remains poorly known. The topic is a key determinant
of changes in how precipitation is partitioned among soil storage, evapotranspiration, and runoff
with implications on ecological response [*Tague and Peng*, 2013; *Trujillo et al.*, 2012] and
regional water resources [*Gleick and Chalecki*, 1999; *Vano et al.*, 2014].





We present a climate sensitivity experiment to investigate how carefully-verified model
simulations of historical snow water equivalent (SWE) and melt rates respond to successively
warmer temperatures that span the range of projected wintertime warming over western North
America for this century [*Van Oldenborgh et al.*, 2013]. A controlled experiment with a
physically based snow model promotes a detailed analysis of the following research questions: 1)
How do SWE and snowmelt rates vary with elevation and how do those gradients vary amongst
dry, average, and wet snow seasons? and 2) How do historical SWE and snowmelt rates respond
to successive degrees of warming?
**2. Methods**
To evaluate the response of SWE and snowmelt dynamics to warmer temperatures, we
conduct a reanalysis of historical snow seasons using the physically based Alpine3D [*Lehning et
al.*, 2006] snow model run at 100 m grid spacing over a mountainous region spanning a 3600-m
elevation gradient in the southern Sierra Nevada, California. Snowpack simulations for three
historical snow seasons were first verified against multi-scale, ground-based observations.
Simulated snowpack characteristics over discrete elevation bands were then examined for their
sensitivity to warmer conditions using a delta-change approach in which observed air
temperature values and the longwave radiative equivalent were augmented by +1°C to +6°C in
+1°C increments. Given the relatively small (< 10%) precipitation changes projected for central
and southern California [*Cayan et al.*, 2008], and a lack of agreement of climate models on the
sign of projected precipitation changes [*Seager et al.*, 2013], the focus of the current study is on
the snowpack response to simulated warming rather than combined changes in temperature and
precipitation. Sensitivity was examined for three historical snow years representative of the





climatological range in snowfall (years with below-average, average, and above-average
snowfall), snow-cover duration, and precipitation timing. The following sub-sections describe
the details of our model experiment, verification, and analysis methods.
**2.1. Study domain**
The study was conducted over a 1648 km$^2$ area encompassing the 1085 km$^2$ Kaweah River basin
on the western slope of the southern Sierra Nevada, California, USA (36.4ºN, 118.6ºW) (Fig. 1).
The elevation of the Kaweah River basin ranges from 250 m to over 3800 m asl. The land-cover
and climate of the domain vary substantially over the full 3633 m elevation range (Fig. 1).
Approximately 98% of the domain is comprised of four land-cover types [*Fry et al.*, 2011]:
conifer forest (58%), shrub (26%), bare soil / rock (10%), and grass / tundra (4%) (Fig. 1). A mix
of grassland, shrub, and oak woodlands characterizes the vegetation of the low elevation foothills
(< 1600 m asl), where mild and wet winters and arid summers characterize the climate and a 660
mm average annual precipitation is rain-dominated [*NPS*, 2017]. At middle elevations (1600 m
to 3000 m asl), mixed conifer forest stands are dominant, including some of the world's only
giant sequoia (*Sequoiadendron giganteum*) groves. The middle elevation climate is cool with
seasonally snow-covered winters and warm, dry summers, and the average annual precipitation
exceeds 1080 mm [*NPS*, 2017]. Forest vegetation of the sub-alpine zone, between 3000 m and
3500 m asl, is sparse and coniferous. Precipitation is not measured at these upper elevations. At
the highest elevations (> 3500 m asl), the land cover is bedrock with sparse alpine vegetation and
snow-cover typically persists from November to July.

The domain includes two research basins: the 7.22 km$^2$ forested Wolverton basin and the

19.1 km$^2$ largely alpine Tokopah basin (Fig. 1). The Wolverton basin is representative of





regional forested mid-elevations. A detailed description of the Wolverton basin instrumentation
is provided in *Musselman et al.* [2012a]. The 19.1 km$^2$ Tokopah basin is representative of small
headwater basins in the southern Sierra Nevada [*Tonnessen*, 1991]. It is instrumented with
numerous meteorological stations and has been the subject of many studies on snow distribution
[*Elder et al.*, 1988; *Girotto et al.*, 2014a; *Jepsen et al.*, 2012; *Marks et al.*, 1992; *Molotch et al.*,
2005] and biogeochemistry [*Perrot et al.*, 2014; *Sickman et al.*, 2003; *Williams and Melack*,
1991]. We use ground-based observations from these research basins to verify the model as
described in Sect. 2.4.
**2.2.   Snow model**

Alpine3D [*Lehning et al.*, 2006] is a land surface model with an emphasis on snow

process representation. It has been used in previous snow process studies [*Bavay et al.*, 2009;
*Magnusson et al.*, 2011; *Michlmayr et al.*, 2008; *Mott et al.*, 2008] and projections of future snow
or runoff [e.g. *Bavay et al.*, 2013; *Bavay et al.*, 2009; *Kobierska et al.*, 2013; *Kobierska et al.*,
2011; *Marty et al.*, 2017]. At the core of Alpine3D is the one-dimensional SNOWPACK model
[*Bartelt and Lehning*, 2002], which has been validated in alpine [e.g. *Etchevers et al.*, 2004] and
forested [e.g. *Rutter et al.*, 2009] environments, including a previous study in the Wolverton
basin using a subset of the forcing and verification data presented herein [*Musselman et al.*,
2012b]. At each model grid cell, mass and energy balance equations for vegetation, snow, and
soil columns are solved with external forcing provided by the atmospheric variables described in
Sect. 2.3.

The bottom (soil) boundary conditions were treated with a constant geothermal heat flux

of 0.06 W m$^{-2}$ applied at the base of a six-layer soil module [see *Musselman et al.*, 2012b for



more information]. In the case of vegetation cover, the surface-atmosphere boundary conditions
were solved for in a single-layer canopy module [see *Musselman et al.*, 2012b]. Wind transport
of snow is not considered in this model implementation. The SNOWPACK model treats the
snowpack as an arbitrary number of layers. New-snow density and snow albedo
parameterizations used in previous studies in the European Alps [*Bavay et al.*, 2013] were found
to work well in the Wolverton basin [*Musselman et al.*, 2012b] and are used in the current study.
Other land-cover parameters such as canopy height and leaf area index were specified according
to land-cover classifications discussed in Sect. 2.3. A simple 1.2°C air temperature threshold was
used to distinguish rain from snow, slightly higher than the 1.0°C value used in *Musselman et al.*
[2012b].
**2.3.  Model input data**
**2.3.1.  Topography and land-cover data**

The elevation and land-cover across the domain were represented at 100 m grid spacing.

Land-cover classification (Fig. 1) was specified from the National Land Cover Database (NLCD)
[*Fry et al.*, 2011]. In addition to the land-cover classes listed in Fig. 1, forest-covered grid cells
were aggregated into coniferous, mixed, and deciduous categories based on the dominant species
within each cell. The NLCD canopy density values, used to parameterize canopy snow
interception and snow surface energy fluxes, were binned from 5% to 85% in 10% intervals.
Grid elements containing vegetation were specified to have an effective leaf area index and
canopy height, respectively, of 0.5 $m^2$ $m^{-2}$ and 1.5 m for shrub/chaparral, 1.2 $m^2$ $m^{-2}$ and 20 m for
deciduous, 2.0 $m^2$ $m^{-2}$ and 30 m for mixed, and 2.7 $m^2$ $m^{-2}$ and 40 m for coniferous forests.





**2.3.2 Meteorological data**
Hourly meteorological observations were available from 19 stations within the domain (Fig. 1
and Table 1). Sixteen stations recorded hourly air temperature and six reported precipitation
(Table 1). The Ash Mountain station at 527 m asl provided the only low elevation precipitation
measurements. The Lower Kaweah station, and the Atwell, Giant Forest, and Bear Trap Meadow
stations are located within a narrow elevation band of 1926 to 2073 m asl (Fig. 1 and Table 1).
Data from a single higher station (Hockett Meadow; 2592 m asl) were not used because of gauge
error for the time period of interest. Precipitation gauge catch efficiency was specified as 0.95 for
rain and 0.6 for snow, using the 1.2°C air temperature threshold as a determinant of precipitation
phase. Incoming shortwave radiation was provided from the Topaz Lake meteorological station
(Fig. 1; Table 1). The direct beam was adjusted for grid cell-specific terrain shading and
elevation dependency and the diffuse component was assumed spatially uniform for each time
step [see *Bavay et al.*, 2013 for details]. The shortwave radiation data are well-correlated with
measurements at middle elevations [*Musselman et al.*, 2012a] and are used to model the full
domain.

The remaining meteorological variables required spatial interpolation from station

locations to all grid cells. Because elevation can have a profound influence on many of the
meteorological variables, several of the interpolation methods used linear elevation trends.
Interpolations were conducted with the data access and pre-processing library MeteoIO [*Bavay*
*and Egger*, 2014] and computed with an Inverse Distance Weighting (IDW) algorithm with
elevation lapse rate adjustments for air temperature, wind speed, and precipitation. Lapse rates
were computed for each hourly time step using a regression technique [*Bavay and Egger*, 2014]
applied to observations from all available stations. If the correlation coefficient was less than 0.6,



then a constant elevation lapse rate of -0.008 °C m$^{-1}$ was used for air temperature and a
standardized elevation trend of 0.0006 m$^{-1}$ was used for precipitation. The incoming longwave
radiation measured at the Topaz Lake station was distributed to all grid cells with a constant
elevation lapse rate of -0.03125 W m$^{-2}$ m$^{-1}$ as in *Bavay et al.* [2013]. Relative humidity was
interpolated as in *Liston and Elder* [2006].
**2.4.   Snow observations and validation data**
**2.4.1 Seasonal basin-scale snow surveys**
Snow surveys were conducted in the two research basins for three snow seasons: 2008, 2009,
and 2010. Three snow surveys of the forested Wolverton basin were conducted each in 2008 and
2009. The survey timing coincided with periods of accumulation (mid-February), maximum
accumulation (mid-March), and melt (late-April). In all three years, early-April surveys of the
alpine Tokopah basin were conducted. In 2009, two additional Tokopah basin surveys captured
accumulation (early-March) and melt (mid-May). Surveys were conducted with graduated
probes to measure snow depth at waypoint locations on a 250 m grid. Surveyors navigated to the
waypoints using Geographic Position System units. At each waypoint, three snow depth
measurements separated by five meters were made along a north-south axis. In total over the
three years, 1,494 waypoints were surveyed. During each survey, snow density was recorded
from snow pits conducted at lower and upper elevations to capture the basin range of snow
density; only one snow pit was dug during the 2010 Tokopah survey. An undisturbed snow face
was excavated to ground and snow density in duplicate columns was measured in 10 cm vertical
intervals by weighing snow samples acquired with a 1000 cm$^{3}$ cutter. In total, 26 snow pits were
measured over the three years. The average snow density at all pits made during a survey was
used to estimate SWE at waypoint locations, which represent the average of three depth




measurements. This approach assumes that basin-scale snow density varies less than snow depth
[*López-Moreno et al.*, 2013].

Simulated SWE at model grid-elements containing waypoint positions are evaluated

against the snow survey values. Three model evaluation metrics are reported. The model bias is
computed as the average difference ('modeled minus measured') of $n$ survey measurements for
each waypoint measurement $SWE_{o_i}$ and corresponding model grid cell $SWE_{m_i}$. The root-mean-
square error (RMSE) is computed as

$$RMSE = \sqrt{\frac{1}{n} \sum_{i=1}^{n} \left(SWE_{m_i} - SWE_{o_i}\right)^2}$$        Eq. (1)

and the normalized mean square error (NMSE) value is computed as

$$NMSE = \frac{\overline{(SWE_m - SWE_o)^2}}{\overline{SWE_m}\ \overline{SWE_o}}$$        Eq. (2)

where the overbars denote the mean over all waypoint locations. The NMSE metric facilitates
model performance comparisons amongst basins, months, and years.
2.4.2 Monthly plot-scale snow surveys
Monthly (1 February – 1 May) manual SWE measurements in the Sierra Nevada are made by the
California Cooperative Snow Survey (CCSS) program to monitor regional water resources.
Seven snow course sites are located within the study domain (Table 1); the sites range in
elevation from 1951 m to 2942 m. At each snow course, linear transects of approximately 10
SWE measurements made with Federal snow tube samplers are averaged to represent the mean
SWE over a distance similar to the 100 m grid cell spacing. The survey measurements thus
provide a SWE estimate that is arguably more representative of the average value within a



corresponding model grid cell than the three point-measurements of the basin-scale surveys or a
single automated SWE station measurement. Modeled SWE values for each survey date at the
grid cells corresponding to each snow course location were evaluated against measured values.
**2.4.3 Automated snow depth sensor network**
In addition to the repeated basin- and plot-scale manual snow surveys, the Wolverton basin
includes a network of 24 ultrasonic snow depth sensors. Four research sites at different
elevations (2253 m, 2300 m, 2620 m, and 2665 m asl) each includes six snow depth sensors and
each site falls within a different 100 m x 100 m model grid cell. The range of snow depth
measured at the six sensors provides a robust estimate of the snow depth, and thus model skill, at
four grid cells spanning slope, aspect, forest density, and elevation in the basin.
**2.4.4 Automated SWE stations**
Daily SWE observations were available from three CCSS automated stations (i.e., snow
'pillows') at middle elevations: Giant Forest (1951 m asl), Big Meadows (2317 m asl), and
Farewell Gap (2896 m) (Table 1 and Fig. 1). Modeled SWE fields were evaluated against these
station observations. The climatological mean SWE record (26 years at Giant Forest and Big
Meadows; 15 years at Farewell Gap) was used to evaluate how the three snow seasons studied
here compare to the long-term average.
**2.5.  Experimental design**
The model was run to simulate seasonal snow dynamics for three reference water years (1
October, 2007 – 30 September, 2010) for which the extensive ground-based observations were
available. Model estimates of snow depth and SWE were evaluated against the observations.



Six warmer temperature scenarios for each of the three reference years were simulated by
increasing the hourly measured air temperature from the 19 regional meteorological stations by
+1°C to +6°C in 1°C increments. The lower (+1°C) and upper (+6°C) limits of simulated
warming correspond to the average winter air temperature increases projected for the year 2100
in western North America in the Representative Concentration Pathway (RCP) emissions
scenarios 2.6 (lowest emissions) and 8.5 (highest emissions), respectively [see top-right panel in
Fig. A1.16 in *Van Oldenborgh et al.*, 2013]. For each warmer temperature scenario (+$n$°C) and
hourly time step $t$, the incoming longwave radiation $LW_{\downarrow t}$ [W m$^2$] measured at the Topaz Lake
station was adjusted for the increase in effective radiative temperature resulting from the warmer
air. The *in-situ* atmospheric emissivity $\epsilon_t$ was estimated from the hourly air temperature $T_{a_t}$
[°C]:

$$\epsilon_t = \frac{LW_{\downarrow t}}{\sigma(T_{a_t}+273.15)^4} \qquad \text{Eq. (3)}$$

where $\sigma$ is the Stefan-Boltzmann constant (5.670373x10$^{-8}$ W m$^{-2}$ K$^{-4}$). The longwave radiation
was adjusted for an effective radiative temperature increase of $n$ [°C] as:

$$LW_{\downarrow t_{(T_a+n)}} = \epsilon_t \sigma(T_{a_t} + 273.15 + n)^4 \qquad \text{Eq. (4)}$$

Relative humidity was held constant to allow water vapor pressure to vary in a manner consistent
with the ideal gas law [*Rasouli et al.*, 2015]. A lack of clear projected wintertime precipitation
response to climate change in the southern Sierra Nevada [see Fig. A1.18 in *Van Oldenborgh et*
*al.*, 2013] prompted our focus on temperature sensitivity rather than a combination of
temperature and precipitation. Observed and adjusted meteorological variables representative of
the warmer scenarios were interpolated to domain grid cells as described in Sect. 2.3.2. The
model was run as in the reference scenarios (Sect. 2.2).



Daily maps of simulated SWE, snow depth, and sublimation were output for each of the
three reference years and six temperature perturbations (21 simulations). For each simulation, we
evaluate the elevational distribution of SWE (mm), daily melt (mm day$^{-1}$), and total annual melt
reported as the depth per unit area (mm per 100 m grid cell) and the total volume (km$^3$). The
daily depletion of SWE, less the daily atmospheric exchange with the snow surface (i.e.,
sublimation and accretion of ice), is a first-order estimate of daily snowmelt (hereafter, snowmelt
rate). The total annual meltwater is then the annual sum of daily snowmelt.
To evaluate how SWE and melt in each scenario varied with elevation, metrics were
averaged or summed into 200 elevation bands, each encompassing ~18 vertical meters, with a
mean of 823 grid cells per elevation band (maximum of 1412). *Rice et al.* [2011] found that
snow disappearance in the Sierra Nevada occurred 20 days later for each 300 m rise in elevation.
The 18 m elevation discretization captures this variability at approximately one day per elevation
band. For each warmer scenario, the total annual meltwater volume is reported as the fraction of
that simulated in the nominal (i.e., unperturbed) case. For all scenarios, we report the annual
meltwater in three ways: the average meltwater volume and melt rate within each elevation band,
the sum of annual meltwater within each elevation band, and the total annual meltwater summed
over the entire model domain. The sensitivity of total domain-wide annual meltwater to
simulated warming is examined with a (linear) regression analysis of the fraction of historical
total meltwater for each warmer scenario of the three years.
To evaluate the effect of simulated warming on melt rates over the elevation profile for
the three years, we report the elevation-specific mean fraction of total annual meltwater produced
at high (≥ 15 mm day$^{-1}$) melt rates, reported as a percent change relative to the nominal case.
The 15 mm day$^{-1}$ threshold was selected as a compromise between the 12.5 mm day$^{-1}$ threshold



above which positive streamflow anomalies were reported by *Barnhart et al.* [2016] and a 20
mm day$^{-1}$ classification of very heavy rainfall [*Klein Tank et al.*, 2009] used by *Musselman et al.*
[2017]. To examine how daily snowmelt rates respond to simulated warming, we present a
quantile analysis of the 25$^{th}$, 50$^{th}$, 75$^{th}$, 90$^{th}$, 95$^{th}$, and 99$^{th}$ percentiles of daily snowmelt rates ≥ 1
mm day$^{-1}$ from the warmer scenarios compared to those from the nominal case. Lastly, we
present an analysis of the meteorological conditions that control the response of snowmelt rates
to successive degrees of simulated warming.
**3. Results**
Maps of simulated SWE on 1 April, 1 May, and 1 June (Fig. 2) highlight seasonal and inter-
annual SWE patterns and illustrate the great variability of SWE with elevation. The lowest
elevations were consistently snow-free during the spring. Middle elevations included a transition
zone from snow-free to seasonally persistent snow-cover; that transition occurred at
progressively higher elevations later in the melt season and occurred earlier (later) in the drier
(wetter) snow years. The upper elevations contained the greatest SWE and most persistent spring
snow-cover (Fig. 2). The three-year observation period captured years with below-average
snowfall (2009; 23% below average SWE; hereafter 'moderately dry year'), average snowfall
(2008; 7% above average SWE; hereafter 'average year'), and above-average snowfall (2010;
54% above average SWE; hereafter 'moderately wet year') as determined from regional
automated SWE records (Fig. 3 and Table S1). The average (hourly) air temperature and
shortwave radiation values measured at the alpine Topaz Lake station in January-February-
March (JFM; the accumulation season) and April-May-June (AMJ; the melt season) provides
more insight into the meteorological differences amongst the three years. The drier and average





years exhibited similar average air temperatures, but the AMJ mean shortwave radiation was
lower in the moderately dry year (Table 2) due to higher spring cloud-cover (see Fig. 6 in
*Musselman et al.* [2012a]). The AMJ period in the moderately wet year was > 2°C colder than
the other years (Table 2) due to a series of large snowfall events in mid-April (Fig. 3) that
prolonged snow-cover (Fig. 2).
**3.1. Model evaluation against observation**

Compared to automated snow pillow SWE measurements, the model performed

favorably (RMSE ≤ 100 mm; bias better than ±85 mm) at all elevations in 2008 and 2010 (Fig.
3). In 2009, the model underestimated SWE compared to measurements made at the two higher
elevation stations, but accurately simulated SWE at the lower Giant Forest station (RMSE = 34
mm; bias = -4 mm) (Fig. 3). The greatest model error occurred at the Big Meadows station (2317
m asl) resulting from a significant underestimation of all snow events and errors were less at the
higher and lower elevation stations (Fig. 3). Despite this underestimation, in general, the SWE
magnitude and date of snow disappearance was well approximated by the model compared to the
automated station measurements.

Compared to the range of snow depth measured by six sensors at each of four sites in the

forested Wolverton basin, the model accurately captured the seasonal snow depth dynamics,
including maximum accumulation, the rate of depletion, and the date of snow disappearance
(Fig. 4; note that simulated snow depth is generally within the measurement envelope). The
general underestimation of SWE in 2009 was not apparent in the verification against the six
automated depth measurements at four sites in the Wolverton basin (Fig. 4).



The early-April surveys of the alpine Tokopah basin show 2009, 2008, and 2010 being

the drier (849±401 mm SWE), average (1000±476 mm SWE), and wetter (1265±310 mm SWE)

snow seasons, respectively (Table S2). Model SWE errors (NMSE) were highest during the melt

season when the measured variability was highest relative to the mean, and lowest during the

accumulation season (Table S2). On average, the forested Wolverton and alpine Tokopah basins

exhibited similar NMSE values of ~ 0.14 at maximum accumulation. In general, the model

tended to overestimate SWE with the exception of the February 2009 Wolverton survey, for

which modeled SWE was negatively biased (Table S2). The survey mean bias values were

typically much less than the standard deviation of the biases.

In general, model SWE errors were lower when evaluated against the CCSS snow course

measurements (Table S3) than the basin-wide survey measurements (Table S2). The large

underestimation of SWE in 2009 seen in the comparison against the automated SWE stations

(Fig. 3) is also seen in comparison to SWE measured at the two lowest elevation snow course

sites (Table S3). Conversely, comparison to the two highest elevation snow course sites indicated

a slight positive model bias in 2009. Overall, the model performed best in regions closest to

precipitation gauges used to force the model; SWE RMSE values were better explained by this

metric than by elevation alone (Fig. S1).

**3.2. Elevation-dependent SWE and snowmelt patterns**

The upper panels of Fig. 5 show the nominal simulations of the daily SWE and melt averaged

along elevation bands for the three years. Persistent seasonal snowpack was simulated >1800 m

asl in all years. Maximum annual SWE increased with elevation (colors in the top row panels of

Fig. 5); however, the date of maximum SWE exhibited a complex relationship with elevation,



snowfall magnitude and timing, and snowpack persistence that all varied amongst years (Fig. 5).
Generally, maximum SWE occurred later with increasing elevation but progressed in a step-wise
manner, often with little change over hundreds of vertical meters interspersed with abrupt jumps
of one to two months (Fig. 5).

Simulated daily melt was episodic in nature with the highest rates (> 35 mm day$^{-1}$; reds in

the bottom panels of Fig. 5) generally confined to elevations > 2000 m asl and the late-spring and
early summer. The highest elevations and years with more/later snow had the highest melt rates.
In all three years, winter melt was generally low (<5 mm day$^{-1}$) with rare, episodic, and more
intense melt events confined to lower elevations (Fig. 5).
**3.3. Elevation-dependent snowpack and snowmelt response to warming**

In the nominal case, the total meltwater volume summed over each elevation band was

consistently greatest between 2500 m and 2800 m asl (see Fig. 6; right panels), corresponding to
the peak in the regional hypsometry (see histograms in Fig. 1). Under the warmer scenarios, the
maximum meltwater volume, inferred from the peaks in Fig. 6, shifts upward in elevation by ~
600 m to the regional treeline (see Fig. 1). This upward elevation shift occurred under +2°C,
+3°C, and +4°C warming for the dry, average and wet snow seasons, respectively. Additional
warming reduced the total melt volume, but did not change the elevation at which the maximum
volume occurred.

Lower and middle elevations were prone to large reductions in the fraction of historical

meltwater volume (see line graphs in Fig. 6). At 2000 m asl, only 50% of the historical water in
the form of snow remained in a +3°C scenario, further reducing to 20% in the +5°C scenario.
Overall, snow at the upper elevations in the moderately dry snow season was more susceptible to



large reductions (Fig. 6). Conversely, upper elevation snowpack during the average and higher
snowfall seasons was more resilient to warming. For example, at 2700 m asl, +1°C warming
reduced annual meltwater volume by 1%, 3%, and 11% in the wetter, average and drier snow
seasons, respectively; those values increased to 7%, 21% and 28% in the +3°C scenario.

Despite elevation-dependent nonlinear meltwater response to warming, the domain-total

meltwater volume exhibited linear response to successive warming. Figure 7 shows linear
regressions fit to the fraction of the nominal-case total meltwater for each scenario and year (see
Table S4). The dry and average years were slightly more susceptible to warming (-10.5% to -
10.8% change per °C) than the wetter year (-9.3% change per °C). Sublimation estimates ranged
from 5% to 9% in the nominal case to 8% to 14% in the +6°C scenario (Table S4).

Warmer temperatures impact not only the total annual meltwater, but also the rate at

which meltwater is produced. Figure 8 shows the fraction of the total meltwater per unit area
over the elevation profile that is produced at high ($\geq$ 15 mm day$^{-1}$) melt rates; the complement of
that fraction occurs at lower (<15 mm day$^{-1}$) rates. Consistently, meltwater production at upper
elevations is dominated by high melt rates, while at lower elevations melt rates are
predominately low. At ~ 2200 m asl, melt in the nominal cases occurred equally at low and high
rates; above this middle elevation zone, melt occurs at high rates ($\geq$ 15 mm day$^{-1}$) and at low
rates (<15 mm day$^{-1}$) below this elevation (see black circle markers in Fig. 8). Warming greatly
decreases the fraction of meltwater produced at high melt rates and increases that produced at
low rates (see lower colored graphs in Fig. 8). As a result, the elevation at which meltwater is
produced equally at low and high rates is pushed upward by ~150 m °C$^{-1}$ (Fig. 8). The greatest
melt rate reductions occur at forested elevations with generally lesser change in alpine areas
above ~ 3300 m asl.





There is a general tendency toward lower snowmelt rates in response to successive
warming with the lower elevations and the year with the most snowfall (and latest storm events)
prone to the greatest reductions (Fig. 9). There are notable exceptions. Extreme melt rates (99th
percentiles; downward-facing triangles in Fig. 9) actually increase (inferred from markers
plotting above the 1:1 line) at elevations > 2800 m asl in all years (top panels) and in the drier
year at all elevations (left panels). To better understand why these extreme melt rates differ in
trend from the lower percentiles, we provide a brief analysis of 2009 extreme melt events. The
analysis is limited to elevations above 2250 m asl where a threshold of 40 mm day$^{-1}$ designates
extreme (99th percentiles) melt rates (see Fig. 9).
In the spring, extreme melt affected a very limited portion of the domain on any given
day (inferred from blue colors on the right in the top panel of Fig. 10), and the spatial extent of
extreme melt generally decreased in response to warming. Conversely, three distinct extreme
melt events on 21 January, 22 February, and 1 March 2009 (arrows in Fig. 10) exhibit large
increases in the fraction of the domain affected, with the January and March events increasing in
spatial extent until +4°C before decreasing with additional warming. The simulated melt events
were not associated with substantial rainfall, but rather cloudy and/or windy conditions with high
longwave radiation that generally occurred under warmer-than-average temperatures in the
nominal case. Measured meteorological conditions for these days are provided in Table 3. These
warm and cloudy winter conditions were insufficient to produce widespread extreme melt in the
nominal case; melt was limited to elevations < 2000 m asl and generally did not exceed the 99th
percentile (Table 3). Additional warming caused extreme rates of melt to occur at increasingly
higher elevations at a time of substantial snow-cover (Fig. 10).
**4. Discussion**



Our results confirm that climate warming will have uneven effects on the California
landscape [*Cayan et al.*, 2008] and that elevation is a critical determinant of snowpack – climate
sensitivity. Despite the simplicity of our climate sensitivity method, the predicted sensitivity of
total snow volume to warming of -9.3% to -10.8% $°C^{-1}$ is consistent with previous studies using
either statistical and dynamical downscaling of GCM output (*Sun et al.* [2016]; -9.3% $°C^{-1}$) or a
simple statistical snow model trained on observations (*Howat and Tulaczyk* [2005]; -10% $°C^{-1}$).
The consistency suggests that these models of varying complexity adequately treat the warming-
induced shift from snowfall to rain. This confirms recent findings by *Schlögl et al.* [2016] that
snow model errors may be negligible when relative climate sensitivity metrics are evaluated.
Further, we show linearity in the sensitivity of domain-wide annual meltwater volume to
successive degrees of warming. The year with the most snowfall, characterized by late snowfall
events and cold spring (AMJ) air temperatures, was slightly more resilient (-9.3% $°C^{-1}$) to
warming than the drier or average snow years. In a study of the sensitivity of snow to warming in
Mediterranean climates, including the Tokopah basin, *López-Moreno et al.* [2017] report that
simulated changes in precipitation magnitude (±20%) did not affect the relative snowpack
climate sensitivity to warming. Thus, snowmelt rates may be more sensitive to changes in the
seasonal timing of precipitation than to changes in precipitation magnitude. This supports the
conclusions of *Cooper et al.* [2016] that record low snowpack years may not serve as appropriate
analogues for the climate sensitivity of snow.
In a warmer climate, shifts from snowfall to rain are likely to combine with shifts in
snowmelt timing to cause earlier water availability relative to the historical period. As a result,
the ephemeral snow zone is expected to progress upward in elevation [*Minder*, 2010] and shift
the areal distribution of SWE toward higher, unmonitored elevations. Indeed, the +3°C scenario



shifted the elevation of maximum annual meltwater volume above that of the highest regional
SWE observing station. The results confirm previous findings in the U.S. Pacific Northwest that
the current observing network design may be insufficient in a warmer world [*Gleason et al.*,
2017]. Warmer temperatures and earlier melt timing [*Stewart et al.*, 2004] also influence the rate
of meltwater production [*Musselman et al.*, 2017], a critical determinant of streamflow [*Barnhart*
*et al.*, 2016], forest carbon uptake [*Winchell et al.*, 2016], and flood hazard [*Hamlet and*
*Lettenmaier*, 2007]. Despite a strong negative relationship between temperature and elevation,
we show a positive relationship between elevation and seasonal snowmelt rates. Compared to
earlier melt at lower elevations, later snowmelt at upper elevations was more rapid due largely to
higher solar insolation coincident with later melt [*Musselman et al.*, 2012a]. Prolonged snow-
cover at upper, compared to lower elevations, and in wetter, compared to drier snow seasons, is
an important factor in interpreting snowmelt temperature sensitivity results.

We show a general tendency toward lower melt rates in response to warming. In contrast

to *Musselman et al.* [2017], which evaluated mean snowmelt response to a single greenhouse gas
emissions scenario at 4 km resolution, we evaluate a range of potential warming, examine the
percentile distribution of snowmelt response, and elucidate the process along elevational
gradients most relevant to basin-wide runoff. This is a critical advancement in understanding
how and where meltwater production is impacted by warming; an evaluation that cannot be
achieved with the type of 'high-resolution' climate modeling used in *Musselman et al.* [2017].
Importantly, we report an emergence (i.e., not present in the historical simulations) and spatial
expansion of extreme winter melt events and, conversely, a decline in extreme melt during
spring. Increases in extreme winter melt occurred under warm and cloudy conditions, and
decreases in extreme spring melt was due to reduced snow-cover persistence. This is an



important new finding with implications on flood hazard and reservoir management. The general
tendency toward slower snowmelt rates and higher extreme values is analogous to the expected
climate change impacts on precipitation, where high-intensity events are expected to increase
despite projected declines in total (e.g., summer) precipitation [*Prein et al.*, 2016; *Trenberth*,
2011].

Increases in extreme winter melt rates, combined with a greater proportion of

precipitation falling as rain could locally increase winter flood risk. Higher winter runoff
complicates reservoir management faced with competing objectives to maintain flood control
storage capacity during winter and to maximize water storage during spring in preparation for the
arid summer. In this context, substantial winter runoff may have to be released downstream
thereby reducing summer water storage required for agriculture, fish and wildlife management,
hydropower production, recreation, water quality and municipal supply [*Lettenmaier et al.*,
1999]. We show that historical extreme melt rates (99$^{th}$ percentiles) impact a relatively limited
area (generally <30% of land area above 2250 m asl) at any given time. This is likely due to
snowpack cold content and/or cool air temperatures limiting melt at upper elevations and low
snow-cover fraction limiting melt at lower elevations. Compared to the historical period,
warming doubles the basin area undergoing extreme melt, and shifts its occurrence from spring
to winter. The increased spatial extent, intensity, and frequency of extreme winter snowmelt
events may have significant implications for antecedent moisture conditions and associated flood
risk.

Snowmelt rates have been mechanistically linked to streamflow production [*Barnhart et*

*al.*, 2016], but less-understood are the potential implications of climate-induced changes in
snowmelt rates on subsurface water storage, evapotranspiration and streamflow response. For





example, recent empirical evidence that a precipitation shift from snow towards rain will lead to
a decrease in streamflow [*Berghuijs et al.*, 2014] lacks definitive causation. Compared to soil,
snow-cover exhibits different water routing mechanisms. For example, lateral downslope flow of
water along snowpack layers has been shown to explain the observed rapid delivery of water to
streams and anomalously high contributions of event water to the hydrograph during rain-on-
snow and snowmelt [*Eiriksson et al.*, 2013]. One hypothesis is that as snow-cover becomes less
persistent in a warmer world, and snowmelt rates decline, this rapid slope-scale redistribution of
water toward stream channels will slow or cease, increasing the soil residence time of water.
Longer soil residence time can increase the partitioning of water to evapotranspiration, and thus
decrease streamflow.
Other empirical and modeling studies have reported declines in summertime streamflow
due to earlier snowmelt runoff and earlier depletion of shallow aquifers [*Huntington and
Niswonger*, 2012; *Luce and Holden*, 2009]. Catchment wetness (i.e., soil moisture content and
shallow groundwater levels) has substantial impact on runoff response in mountainous areas with
distinct thresholds determining relationships amongst wetness, streamflow, and contributing area
[*Penna et al.*, 2011]; behavior controlled by soil type, subsurface storage capacity, and climate.
These factors are also important drivers of evapotranspiration [*Christensen et al.*, 2008;
*Lundquist and Loheide*, 2011] and the regional variability of hydrologic sensitivity to climate
change [*Tague et al.*, 2008]. In this regard, percentage reductions in future streamflow may be
more substantial than the meltwater reductions reported here because slower snowmelt is less
efficient at generating streamflow.
Improved model error characterization for the baseline (nominal) years is a critical step
toward informed interpretation of the results of our climate change sensitivity analysis. While



snow model errors may be negligible when relative climate sensitivity metrics are evaluated
[*Schlögl et al.*, 2016], runoff simulations require accurate representation of snowpack volume
and melt rates. Simulated snow depth values were within the range of observations from
automated sensors at four sites spanning elevation, forest density, slope and aspect. This
verification provides confidence in the model to capture accumulation, melt rates, and the date of
snow disappearance across spatial and temporal scales. Notwithstanding, there are inherent
strengths and weaknesses of the different validation data sets. For example, the basin-scale
survey design samples only three snow depth measurement points within a given grid cell.
Similarly, the automated stations only sample a single point. The degree to which these point
samples represent the average value over an area consistent with the model grid scale is a source
of inherent discrepancy between models and observations, independent of model skill [*Trujillo*
*and Lehning*, 2015]. Overall, the model performed best in regions closest to precipitation gauges
used to force the model (Fig. S1) and tended to slightly overestimate SWE at upper elevations
(Table S3).

Our assumption of a uniform temperature perturbation does not consider changes in

climate dynamics at diurnal (e.g., nighttime vs. daytime temperature changes), synoptic (e.g.,
number of cool vs. warm days), or seasonal (e.g., winter vs. spring temperature changes) scales.
These interactions may be best characterized using GCM output dynamically downscaled to fine-
resolutions with regional climate models [e.g., *Liu et al.*, 2016; *Sun et al.*, 2016] or within a
delta-change approach that considers the range of uncertainties in the climate change signal of
different emissions scenarios [e.g., *Marty et al.*, 2017]. By not addressing the snow-albedo
feedback between snow-cover depletion and warmer temperatures [*Letcher and Minder*, 2015;
*Pepin and Lundquist*, 2008], it is possible that we underestimate regional air temperature



changes toward the end of the melt season in the warmer scenarios. Such negative temperature
biases would cause underestimation of the snow depletion rate and, ultimately, the snowpack
sensitivity to warming. However, these biases may be partially mitigated by our assumption that
the winter and spring, and nighttime and daytime, air temperatures warm uniformly.
Sublimation estimates of 5% to 9% in the nominal case to 8% to 14% in the +6°C
scenario (Table S4) are on the lower- to middle-end of the reported regional values of 2% to 3%
[*West and Knoerr*, 1959] to 20% [*Marks and Dozier*, 1992]. The large range highlights
challenges and disparities in measuring [e.g., *Molotch et al.*, 2007; *Sexstone et al.*, 2016] and
modeling [*Etchevers et al.*, 2004] turbulent exchange, which are further compounded in
mountainous terrain due to the challenges of windflow simulation [*Musselman et al.*, 2015]. By
not considering blowing snow and subsequent sublimation losses (i.e., overestimating alpine
snowpack), we may further underestimate snowpack sensitivity to warming.
In light of the potential errors discussed above, our results should be considered
somewhat conservative. Longer-term snow and runoff simulations at scales sufficient to resolve
mountain climate elevation gradients are needed both as reanalysis to understand historical
conditions [e.g., snow reanalysis by *Margulis et al.*, 2016], and forced by large suites of future
climate scenarios [e.g., *Eyring et al.*, 2016] that dynamically resolve different model realizations
of climate response to different greenhouse gas emissions scenarios. Such efforts will best
inform, and constrain the uncertainty of, potential impacts of climate change on flood risk and
water availability. Toward this goal, our work makes inroads to quantify how snowpack and melt
dynamics respond to incremental warming over an elevation profile characteristic of a foothills-
to-headwaters mountain front. The results offer insight into the sensitivity of snow water



resources to climate change in the Sierra Nevada, California, with implications for other regions
as well.

## 5. Conclusions

We present a climate sensitivity experiment to investigate how historical snow water resources
and melt rates respond to successively warmer temperatures over a large elevation gradient in the
southern Sierra Nevada, California. Good agreement between simulations and an unprecedented
array of ground-based observations of SWE (RMSE ≤ 100 mm; bias better than ±85 mm) and
snow depth (within multi-sensor range) is shown. Three primary findings emerge from the
simulations. First, the sensitivity of total snow-water volume to warming is -9.3% to -10.8% per
°C. The snow season characterized by above-average snowfall and cold spring storm events was
most resilient to warming; however, it also exhibited the greatest shift toward slower melt. Thus,
snowmelt rates may be more sensitive to changes in the seasonal timing of precipitation than to
changes in precipitation magnitude. Second, the middle elevations, which are dominated by
forest cover and comprise a disproportionately large basin area, exhibit the greatest snowpack
reductions and the largest shift toward slower snowmelt. Hence, warming-related impacts on
runoff production and ecosystem function may be particularly acute in these areas. Third,
increases in the frequency, intensity, and spatial extent of extreme winter melt events occur with
successive warming. Warming-induced extreme (winter) melt impacts an area nearly twice as
large as that simulated at any time in the historical period. The changes in extreme snowmelt
events have implications for antecedent moisture conditions and associated flood risk. When
considered together, the elevation-dependent climate sensitivity of snowmelt revealed herein has
broad implications for water supply monitoring, streamflow production, flood control, and
ecosystem function in a warmer world.



**Acknowledgements**
The authors thank Sequoia National Park for support of research efforts. Financial support was
provided by the National Science Foundation grants EAR-1032295, EAR-1032308, and EAR-
1246473, the Southern Sierra Critical Zone Observatory (EAR-0725097), a Major Research
Instrumentation grant (EAR-0619947), and the Mountain Research Initiative. The first author
was supported by a National Aeronautics and Space Administration (NASA) Earth System
Science Fellowship. R. Bales and P. Kirchner supported hydrometeorological infrastructure in
the Wolverton basin. J. Sickman and J. Melack provided solar radiation and snow survey data
from the Tokopah basin. Alpine3D is provided the WSL Swiss Federal Institute for Snow and
Avalanche Research SLF (online: https://models.slf.ch/p/alpine3d/downloads/). Special thanks
goes to M. Lehning and M. Bavay. Model forcing data are freely available online from the
agencies listed in Table 1. Land-cover and validation data are either available online from
sources referenced in the text, or are otherwise provided in the figures, tables, and supplements.
The authors are grateful to everyone who provided field assistance including: K. Skeen, S.
Roberts, B. Forman, D. Perrot, E. Trujillo, L. Meromy, M. Girotto, A. Kahl, K. Ritger, N. Bair,
D. Berisford, A. Kinoshita, and M. Cooper.



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



**Tables**
Table 1. Meteorological station and snow measurement details. Station numbers are
ranked by station elevation and correspond to those mapped in Fig. 1. The variables
measured at each location are listed: air temperature (Ta), relative humidity (RH), wind
speed (ws), precipitation (ppt), snow water equivalent (SWE), and snow depth (depth).

| # | Station name | Elev., m | Measured variables* | Operating agency |
|---|---|---|---|---|
| *Automated met. stations* | | | | |
| 1 | D0117 | 263 | Ta, RH, ws | APRSWXNET |
| 2 | C4177 | 378 | Ta, RH, ws | APRSWXNET |
| 3 | Ash Mountain | 527 | Ta, RH, ws, ppt | NPS |
| 4 | Shadequarter | 1323 | Ta, RH, ws | CDF |
| 5 | Wolverton | 1598 | Ta, RH, ws | NPS |
| 6 | Lower Kaweah | 1926 | Ta, RH, ws, ppt | NPS |
| 7 | Atwell | 1951 | ppt | USACE |
| 8 | Case Mountain | 1967 | Ta, RH, ws | BLM |
| 9 | Giant Forest | 2027 | Ta, ppt | USACE |
| 10 | Bear Trap Meadow | 2073 | ppt | USACE |
| 11 | Wolverton Meadow | 2229 | Ta, RH, ws | SNRI |
| 12 | Park Ridge | 2299 | Ta, RH, ws | NPS |
| 13 | Hockett Meadows | 2592 | ppt | USACE |
| 14 | Marble Fork | 2626 | Ta | ERI |
| 15 | Panther Meadow | 2640 | Ta, RH, ws | SNRI |
| 16 | Emerald Lake | 2835 | Ta, RH, ws | ERI |
| 17 | Farewell Gap | 2896 | Ta | USACE |
| 18 | Topaz Lake | 3232 | Ta, RH, ws, SW, LW | ERI |
| 19 | M3 | 3288 | Ta, RH, ws | ERI |
| *Automated snow stations* | | | | |
| 1 | Giant Forest | 1951 | SWE | USACE |
| 2 | Big Meadows | 2317 | SWE | USACE |
| 3 | Farewell Gap | 2896 | SWE, depth | USACE |
| *Monthly snow courses* | | | | |
| 1 | Giant Forest | 1951 | SWE, depth | NPS |
| 2 | Big Meadows | 2317 | SWE, depth | CADWP |
| 3 | Mineral King | 2439 | SWE, depth | NPS |
| 4 | Hockett Meadow | 2592 | SWE, depth | NPS |
| 5 | Panther Meadow | 2622 | SWE, depth | NPS |
| 6 | Rowell Meadow | 2698 | SWE, depth | KRWA |
| 7 | Scenic Meadow | 2942 | SWE, depth | KRWA |

*Meteorological variables used in this study.
APRSWXNET: Automatic Position Reporting System as a Weather NETwork
NPS: National Park Service (Sequoia and Kings Canyon National Parks)
CDF: California Department of Forestry
USACE: United States Army Corps of Engineers
BLM: Bureau of Land Management
SNRI: Sierra Nevada Research Institute, University of California Merced
ERI: Earth Research Institute, University of California Santa Barbara
CADWP: California Department of Water and Power
KRWA: Kaweah River Water Association



Table 2. Average (hourly) air temperature and shortwave radiation values measured at the alpine
Topaz Lake meteorological station in the Tokopah Basin for JFM and AMJ of the moderately
dry year (2009), near-average year (2008) and moderately wet year (2010).

|  | Air temperature, °C | | Shortwave, W m$^{-2}$ | |
|---|---|---|---|---|
|  | JFM | AMJ | JFM | AMJ |
| 2009 | -3.2 | 3.3 | 163 | 279 |
| 2008 | -3.6 | 3.4 | 166 | 317 |
| 2010 | -4.0 | 1.3 | 152 | 306 |






Table 3. Mean daily values of hourly measured meteorological variables (nominal mean) during
the three mid-winter melt events in 2009 (see Fig. 10) compared to the average conditions
measured at eight stations > 2250 m asl computed on 11-days centered on the event dates,
averaged over the three years of the study. Precipitation is reported as the daily sum of measured
values. Melt rates simulated in the nominal case are reported as the mean value computed over
all grid elements > 2250 m asl and the maximum value over the full domain with the
corresponding elevation.

| Met. variable | Jan. 21 | Feb. 22 | Mar. 1 |
|---|---|---|---|
| Air temp., °C | 2.8 / -0.5 | -0.7 / 0.6 | 4.4 / 1.0 |
| Shortwave, W m$^{-2}$ | 57 / 96 | 83 / 152 | 163 / 176 |
| Longwave, W m$^{-2}$ | 292 / 232 | 297 / 226 | 266 / 217 |
| Wind, m s$^{-1}$ | 4.0 / 4.3 | 4.6 / 4.0 | 7.2 / 4.4 |
| Precipitation, mm | 0.0 | 4.3 | 0.0 |
| Mean melt rate, mm d$^{-1}$ nom. sim. (>2250 m) | 6.5 | 1.5 | 4.7 |
| Max. melt rate, mm d$^{-1}$ nom. sim. (elev., m) | 30.6 (1897) | 28.3 (1586) | 44.0 (1741) |






**Figures**

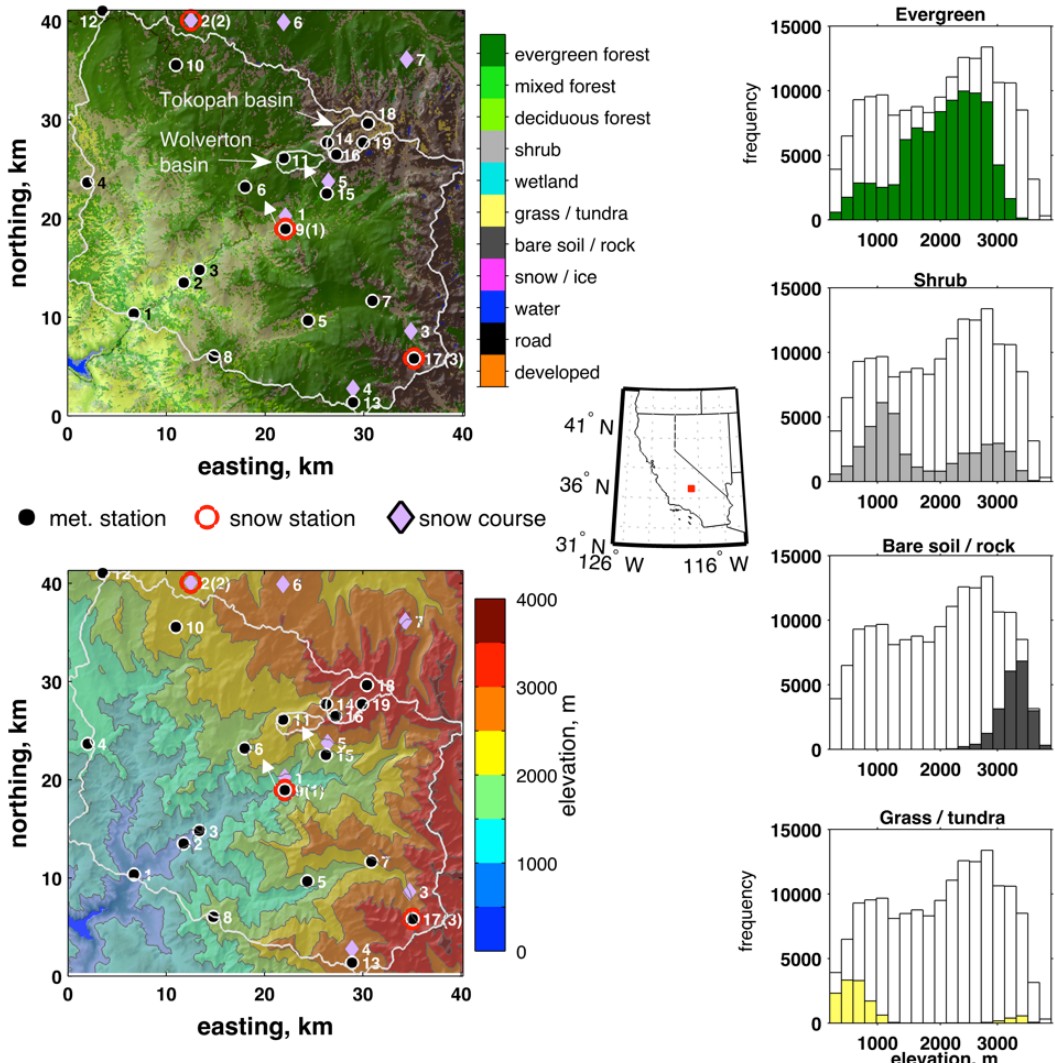

Figure 1: The elevation and land cover distribution of the model domain encompassing the
Kaweah River basin (outlined) on the western side of the southern Sierra Nevada, California.
Locations of the forested Wolverton and largely alpine Tokopah research basins are indicated.
The locations of 19 automated meteorological stations (filled circle markers), three automated
snow stations (red circles), and, seven monthly snow survey transects (diamond markers) are
shown. Station numbers, ranked by elevation, correspond to those in Table 1. The histograms
illustrate the elevation distribution of the four primary land cover types (colored bars) relative to
the elevation of the model domain (empty bars).





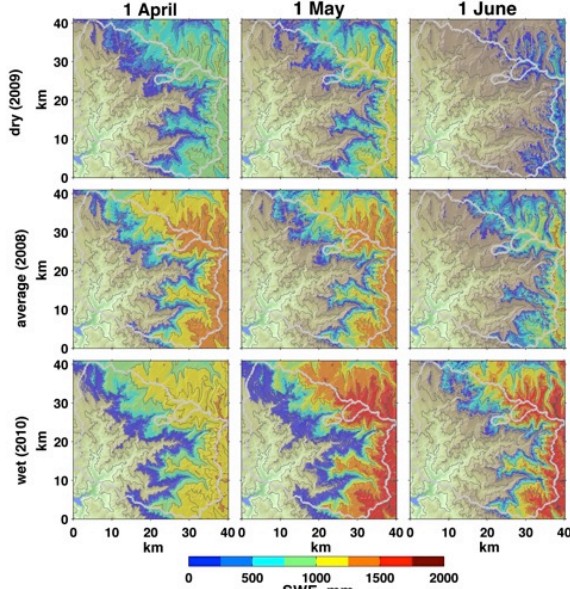

Figure 2: Simulated SWE over the greater Kaweah River basin on the first of April (left panel column), May (center panel column), and June (right panel column) for a moderately dry water year (2009; top panel row), near-climatological-average water year (2008; middle panel row), and a moderately wet water year (2010; bottom panel row).



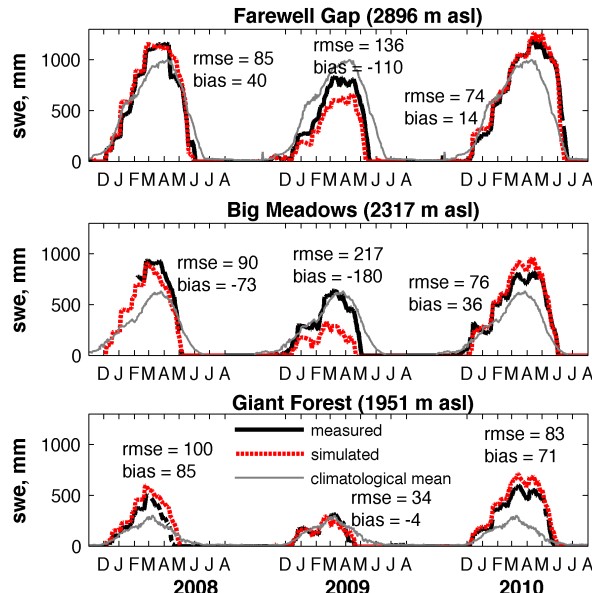


Figure 3: Measured and simulated SWE at the three automated snow stations spanning the
middle elevations of the greater Kaweah River basin. The error metrics RMSE and bias, in
millimeters, are provided for each station-year. The thin gray line indicates the long-term
climatological mean SWE based on 26-years of data (1988 – 2014) collected at the Giant Forest
and Big Meadows stations and a 15-year record (2000 – 2014) at the Farewell Gap station.




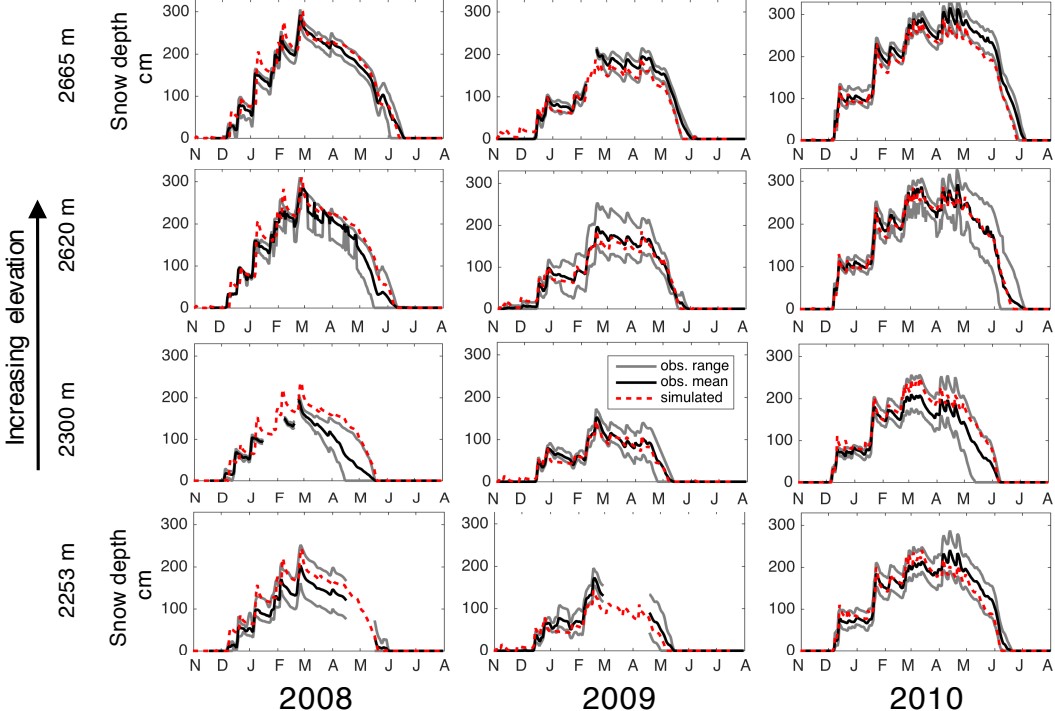

Figure 4: Comparison of three years (panel columns) of daily (x-axes) simulated (red lines) snow
depth and the six-sensor observed range (gray lines) and mean (bold lines) snow depth measured
by automated sensors at four research sites (panel rows) at different elevations in the Wolverton
basin.



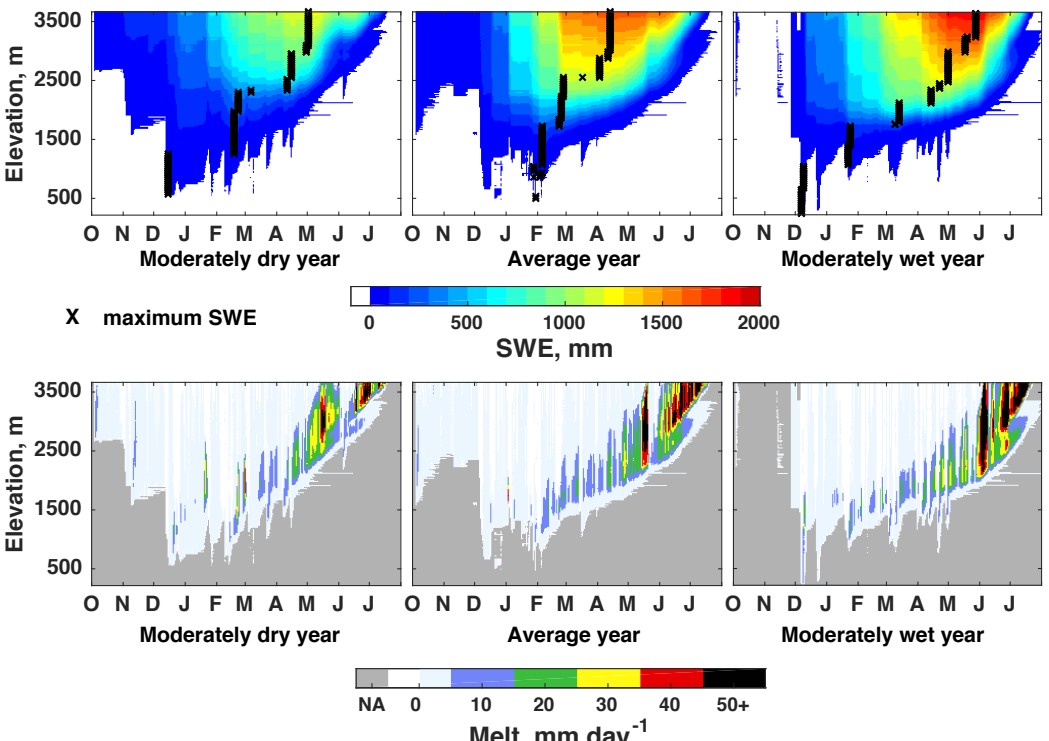


Figure 5: Distribution of (top panels) SWE and (bottom panels) daily melt by elevation (mean
values within 18 m elevation bins; y-axes) and time (daily; x-axes) for a moderately dry (2009;
left column panels), near-average (2008; center column panels), and moderately wet (2010; right
column panels) snow season. The grey color in the lower panels indicates times when there is no
snow to melt (NA). The elevation-specific dates of maximum SWE are indicated.



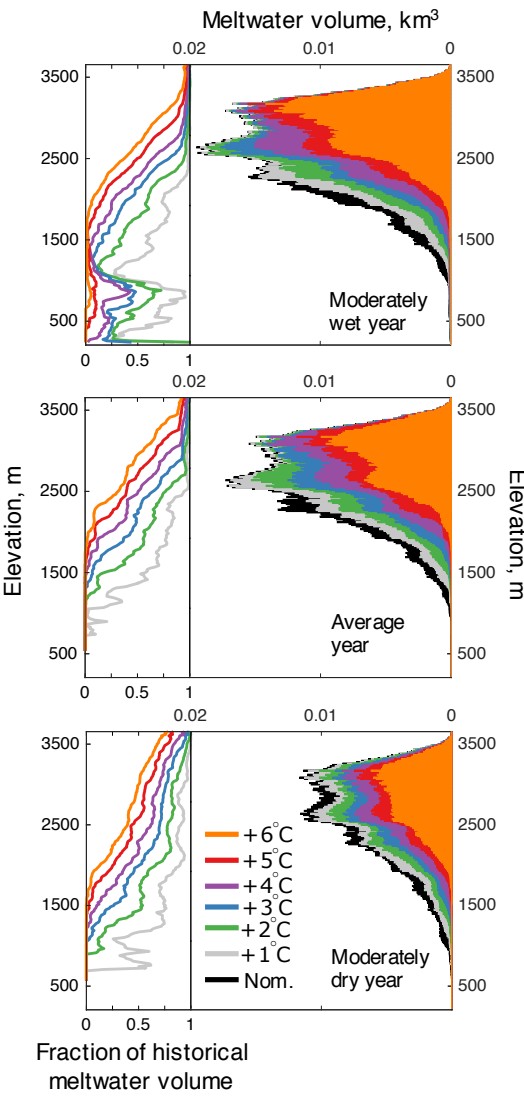

Figure 6: The elevation distribution (y-axes) of (right bar graphs) simulated annual meltwater
volume and (line graphs) the fraction of that historical meltwater for each warmer scenario
(colors; see legend) for the (top) moderately wet, (middle) average, and (bottom) moderately dry
snow seasons. The total meltwater was summed within the same elevation bins used in Fig. 5.





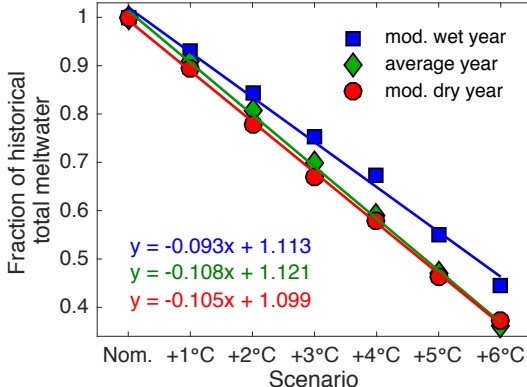

Figure 7: The fraction of simulated domain-wide historical meltwater (y-axis), relative to the
nominal case, for each warmer temperature scenario (x-axis) for the three years (marker type and
color). The colored lines and associated regression equations show linear fits to the data. For
each year, the $R^2$ value was $> 0.99$ and the p-value was $\ll$ 1e-6.





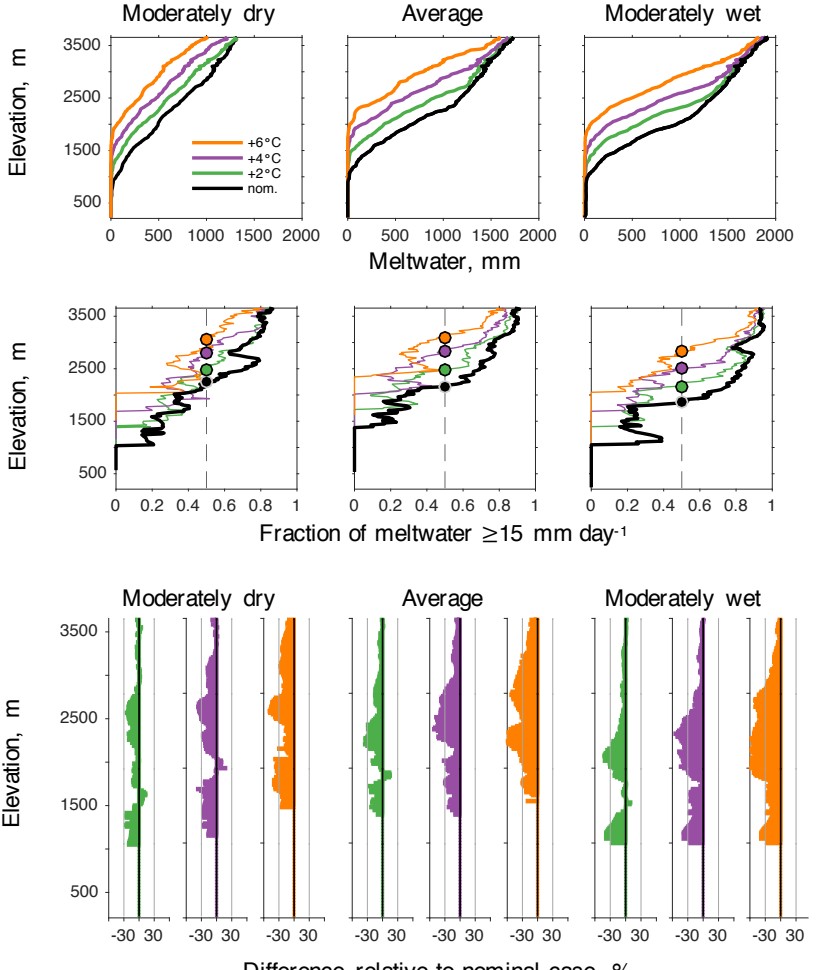


Figure 8: The elevation distribution (y-axes) of (top row of panels) the average total depth of
annual meltwater (x-axes) simulated for the nominal case (black lines) and select perturbed
temperature scenarios (colored lines), and (second row of panels) the fraction of annual
meltwater produced at snowmelt rates $\geq$15 mm day[-1]. The colored circles indicate elevations at
which simulated melt occurs equally at rates $\geq$15 mm day[-1] and <15 mm day[-1]. The lower panels
of colored graphs show the differences from the nominal case, reported in percent of annual
meltwater, produced at snowmelt rates $\geq$15 mm day[-1] for the three select scenarios. Results are
shown for the moderately dry (2009; left column of plots), near-average (2008; middle column of
plots), and moderately wet (2010; right column of plots) snow seasons.





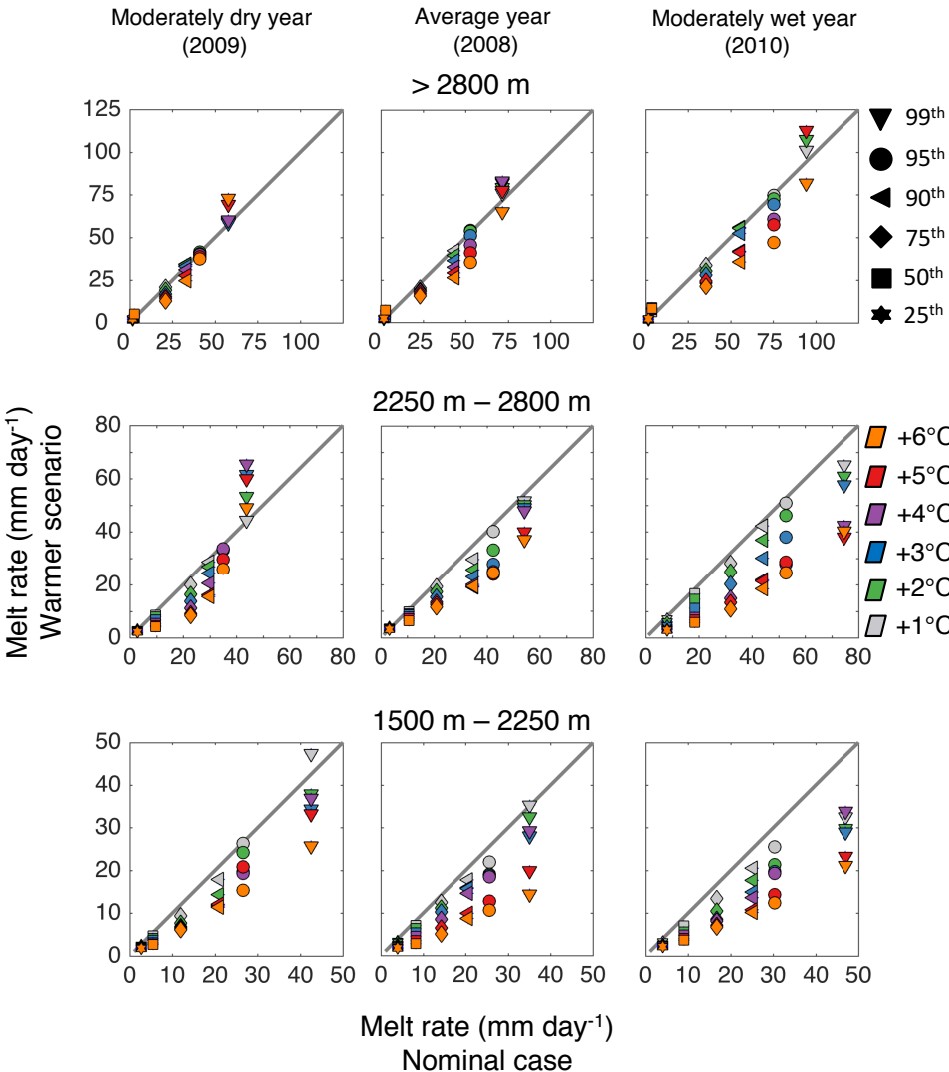

Figure 9: Quantile plots of simulated melt rates for the nominal (x-axes) and warmer scenarios
(y-axes) for model grid cells characterized as high elevation (> 2800 m; top row of panels),
middle elevation (2250 m – 2800 m; middle row of panels) and lower elevation (1500 m – 2250
m) regions for the moderately dry year (left column), average year (middle column) and
moderately wet year (right column). Marker colors correspond to the six different temperature
perturbations. Plotted in each graph are the 25th, 50th, 75th, 90th, 95th, and 99th percentiles (marker
shapes) of daily snowmelt rates ≥ 1 mm day$^{-1}$ for all grid cells within each water year and
elevation range. The 1:1 lines are plotted for reference.



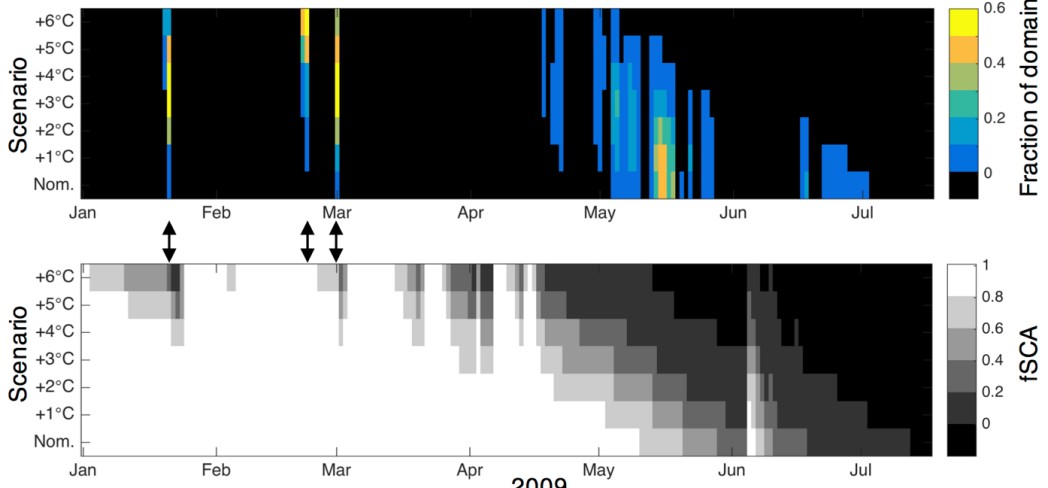


Figure 10: Daily extreme snowmelt in 2009 (melt rates > 40 mm day$^{-1}$ at model grid cells > 2250
m asl, corresponding to extreme melt rates [≥ 99$^{th}$ percentile]; see Fig. 8) as simulated by the
nominal (Nom.) and six perturbed temperature scenarios (y-axes) shown as the (top panel)
fraction of the area undergoing extreme melt. The lower panel shows the fraction of snow-
covered area (fSCA) for the same time period and domain. Arrows indicate (winter) melt events
(see Table 3 for meteorological conditions and averages).