# Peer review of "Snowmelt response to simulated warming across a large elevation"

_The Cryosphere, 2017_

## Referee Comment (RC1) · E. Sproles (Referee) · 3 Sep 2017

Reviewer Comments for:

*Snowmelt response to simulated warming across a large elevation gradient, southern Sierra Nevada, California*

By Musselman et al.

Submitted to The Cryosphere

The authors provide a new approach to addressing projected climate impacts on mountain snowpack by focusing on snowmelt. Most previous studies have focused on snowpack. Their perspective is relevant, especially for water management.

Overall the structure of the paper was well organized and presented, and the conceptual design and research is worthy of publication. Before this research is publishable I have one major concern regarding the lack of detail and specifics of the modeling for meteorological conditions and validation of snowpack.

From the paper I inferred that the authors modeled snowpack using all of the available meteorological stations, but did not validate the meteorological outputs. For example, the authors state *"Interpolations were conducted with the data access and pre-processing library MeteoIO [Bavay and Egger, 2014] and computed with an Inverse Distance Weighting (IDW) algorithm with elevation lapse rate adjustments for air temperature, wind speed, and precipitation."*

The authors had 19 stations of meteorological forcing data. But were any used to assess the meteorological outputs before calibrating snowpack?

If not the authors calibrated a model to three snow years and then applied projected increased temperatures. If this is the case, their approach is problematic. It is critical to ensure that you are getting modelled results correct for the right reasons (Kirchner, 2006). Modeling snowpack without ensuring that you have the underlying meteorological conditions correct is problematic, because any modeled error propagates through the climate projections. It is pertinent to get the right answers for the right reasons and demonstrate this to the reader.

In my opinion this paper should not be published until the authors clarify their approach or remove some of the forcing stations and use them to validate the meteorological outputs.

Other specific comments:
Line 1: The abstract is very well written, with the exception of the first sentence. I did not understand your point until I read the sentence three times. Please consider restructuring it so that there is little to no ambiguity.

Lines 90 – 93: I like your description of why you chose only to change changes in temperature.

Line 143: You describe that the model has an arbitrary number of layers. I found myself wanting more information about how the number of layers is/was decided. By the modeler or the model?

Lines 242-243: How were the automated SWE measurements evaluated? This should be provided in the methods.

Lines 281 – 282: This is a very informative way to organize the elevation bands!

Lines 327 – 329: Please provide some of the result values for Snow Disappearance Date.

Lines 341 – 346: There are a lot of generalities (i.e. in general is used two times in three sentences). It would be informative to have to have detailed values and let the reader decide in general or not. In its present form, the reader simply has to assume what the authors are telling them with minimal supporting data/evidence.

Lines 359 – 361: I did not see the step-wise jump that you all are referring to here. I fully assume it is my error, but it might be helpful to annotate the jumps for your readers.

Lines 368 – 375: This is great data, and well organized.

Lines 445 – 455: This is an important policy/infrastructure component of this work. Consider adding an even more bold statement here. Something along the lines of: "The shift of SWE towards higher unmonitored elevations highlights the need for expanding the existing monitoring network to better manage water resources. For example, in the extremely low snow year of 2015, 1 March 47% of snow monitoring sites in the Willamette River basin, Oregon registered zero SWE, while snow was still present at higher elevations. (Sproles et al., 2017)".

Use the citation or not, but I recommend a stronger policy statement regarding the absence of SWE monitoring at higher elevations, and how the addition of these sites would be critical to monitor snowmelt response to warmer temperatures and for water resource management.  Your efforts demonstrate why additional monitoring is important.

Lines 481 – 483: This statement is an important discussion point that could be made stronger by adding a more recent citation and examples from the winters of 2014 and 2015.

Lines 519 – 520: Throughout the paper I was expecting snowmelt lysimeters to be included, as either data or as discussion topic. I found the absence of any discussion of

lysimeters in a paper focused on snowmelt to be incomplete. Are there no lysimeters in the study area? If not would this also be an infrastructure recommendation?

At the minimum, I would suggest a brief explanation of how lysimeters could or could not improve the scientific understanding of snowmelt processes.

Lines 529 – 531: This is a pretty critical point, especially since one of your primary findings is that the higher elevation snowpack will be more important under warmer conditions. The fact that the model performs better near forcing stations further supports my concerns modeled temperature and precipitation data not being calibrated/validated. This is an important component of modeling distributed snowpack. Doing so could perhaps improve model performance at higher elevations.

Figure 2: Your color ramp for snow is counterintuitive. Red usually represents warning/drier conditions, but here it represents more snow. Additionally, if you are displaying sequential data it should be on a single color ramp.

Light, Adam, and Patrick J. Bartlein. "The end of the rainbow? Color schemes for improved data graphics." *Eos* 85.40 (2004): 385-391.

Also, http://colorbrewer2.org/, has color schemes that work really well. I believe there is a Matlab function for Colorbrewer as well.

https://www.mathworks.com/matlabcentral/fileexchange/34087-cbrewer---colorbrewer-schemes-for-matlab

Figure 5: Again, colors are counterintuitive and should be redesigned.

Figures 6, 7, and 10 are really informative. They provide a lot of information!

---

## Referee Comment (RC2) · B. Henn (Referee) · 8 Sep 2017

General Comments

This study examines the effect of projected climate warming on snow accumulation and melt rates along an elevation gradient and between wet and dry years. The authors use detailed snow and meteorology observations from Sequoia National Park to validate and drive physically based snow model simulations of historic and projected snowpack under warming. The simulations show that historic conditions are reproduced for three years of data with acceptable agreement to observations. They then show sensitivity of snow volumes and melt rates with elevation and wet/dry years, finding that snow

volumes decline with warming at about 10%/degC overall with greater losses in the mid elevation coniferous forests, and that melt rates decline in areas of snow loss as melt now occurs earlier in the year. They also argue that extreme high melt rates increase under warming based on one of the three simulated years in which mid-winter melt events occurred more extensively.

The study reinforces previous work that shifts from snow to rain under warming will result in lower snowpacks in mid-elevation areas, and that the lower, more ephemeral snowpacks will melt more slowly and earlier in the winter. The elevation gradient and three years of varying precipitation are helpful in visualizing those effects across those important dimensions of the mountain hydroclimate, making the study a useful contribution. The authors' claim that extreme melt rates (defined as high-quantile rates under the historic scenario) may increase under warming is novel and has flood risk implications. This claim may need to be phrased more carefully as the majority of the results presented show melt rates decreasing under almost all warming scenarios, and it also seems possible that the increases in melt rates reported are sensitive to the assumptions of the warming perturbations in this particular study. The authors should perhaps more clearly state some of these caveats.

The paper is well written and clearly organized and has high-quality figures.

Specific Comments

L254-262: The section on how longwave radiation was calculated under the perturbation scenarios is clear in terms of how it was done, but would benefit from more conceptual explanation about why this method is appropriate. For example, this method assumes that both RH and emissivity of the atmosphere do not change under the warming scenarios. While there is conceptual evidence to support (fairly) similar RH in a warmer climate, emissivity has a dependence on temperature (Flerchinger, G. N., W. Xaio, D. Marks, T. J. Sauer, and Q. Yu (2009), Water Resour. Res., doi:10.1029/2008WR007394.) Given that the authors cite midwinter melt rates as a

key finding later in the study, and longwave has been implicated in driving midwinter melt (Lundquist, J. D., Dickerson-Lange, S. E., Lutz, J. a., & Cristea, N. C. (2013). Water Resources Research, http://doi.org/10.1002/wrcr.20504), justifying the perturbation assumptions around longwave and turbulent energy fluxes might benefit the paper.

L295-297: The explanation of the quantile analysis of melt rates could perhaps be better. If I am understanding correctly, the 99th percentile (for example) melt rates (over the whole spatial domain and year?) are calculated for the nominal case, and then this is repeated for the warming cases and the melt rates are compared?

L376-389: Somewhere in the paper, discussion about why drier years were more susceptible snowpack loss to warming than wetter years might be warranted. Given that precipitation is fixed, were snow accumulations reduced more in the dry year because those storms were warmer (rain-snow level closer to mean domain elevation) than in wet years? i.e., is this a general finding or something specific to the storms in those years?

L405-408: Perhaps I am misunderstanding Figure 9, but the statement that

"Extreme melt rates (99th percentiles; downward-facing triangles in Fig. 9) actually increase (inferred from markers plotting above the 1:1 line) at elevations > 2800 m asl in all years (top panels) and in the drier year at all elevations (left panels)"

doesn't quite seem to follow the data - most of the warming scenarios showed did not show increasing 99th percentile rates for the dry year at low elevations, nor did several of the scenarios for each year the high elevation zone. It is interesting that some extreme melt rates did increase, but focusing only on those scenarios that increased might overstate the robustness of this finding.

L425: The discussion section could benefit from better organization and potentially using subsections to divide it among topics. The ordering of this section (discussing results, then flood and soil moisture implications, then caveats and other issues) seemed

a bit meandering for me as a reader (implications for streamflow and soil moisture, processes which are not tested in this paper, seem like they should go last, for example).

L545: The mechanisms behind the reduction in snowpack with warming seem like they deserve greater discussion here. If I am understanding the experiment, there are only two mechanisms by which warming reduced meltwater volumes: precipitation falling as rain instead of snow, and increased sublimation. How important are these relative to each other in reducing meltwater volumes? Is increased sublimation significant at all or is it entirely the shift to rain? Some discussion of these mechanisms behind the results might be helpful to placing them in physical context.

Technical Corrections

Figure 1: Having repeated numbering of different types of stations is confusing - consider different ways of numbering sites.

---

## Author Comment (AC1) · 16 Oct 2017

Reviewer Comments for:
Snowmelt response to simulated warming across a large elevation gradient, southern
Sierra Nevada, California
By Musselman et al.
Submitted to The Cryosphere

The authors provide a new approach to addressing projected climate impacts on mountain snowpack by focusing on snowmelt. Most previous studies have focused on snowpack. Their perspective is relevant, especially for water management.

Overall the structure of the paper was well organized and presented, and the conceptual design and research is worthy of publication. Before this research is publishable I have one major concern regarding the lack of detail and specifics of the modeling for meteorological conditions and validation of snowpack.

Thank you for your review. We appreciate the recognition of our paper's contribution and scientific relevance. Your support of the ultimate publication of the paper is encouraging. We have addressed suggestions and comments below. Our responses are provided in blue font. References to line numbers refer to those of the revised manuscript.

From the paper I inferred that the authors modeled snowpack using all of the available meteorological stations, but did not validate the meteorological outputs. For example, the authors state "Interpolations were conducted with the data access and pre-processing library MeteoIO [Bavay and Egger, 2014] and computed with an Inverse Distance Weighting (IDW) algorithm with elevation lapse rate adjustments for air temperature, wind speed, and precipitation."

The authors had 19 stations of meteorological forcing data. But were any used to assess the meteorological outputs before calibrating snowpack?

If not the authors calibrated a model to three snow years and then applied projected increased temperatures. If this is the case, their approach is problematic. It is critical to ensure that you are getting modelled results correct for the right reasons (Kirchner, 2006). Modeling snowpack without ensuring that you have the underlying meteorological conditions correct is problematic, because any modeled error propagates through the climate projections. It is pertinent to get the right answers for the right reasons and demonstrate this to the reader.

In my opinion this paper should not be published until the authors clarify their approach or remove some of the forcing stations and use them to validate the meteorological outputs.

We clarify our modeling and validation approaches with three main points A), B) and C). Associated changes to the manuscript are detailed below each point.

A)      Unlike a temperature index snowmelt model or a hydrologic model, energy balance snowmelt models are not commonly calibrated. As such, no calibration of the meteorological interpolation procedures or snow model was performed. This approach

relies on the idea that a model based on good understanding of the physical principles and basin characteristics, with an appropriate structure, spatial resolution and parameter selection, should have a good chance of simulating the hydrological cycle including snow accumulation (Pomeroy et al., 2007).

The use of an uncalibrated physically based model requires carefully selected parameters and verification data to ensure model accuracy and characterize model error. This lends confidence that we get the "right answers for the right reasons". For example, "The combination of targeted field observations and uncalibrated physically-based model diagnosis can provide for rapid advances in the understanding of hydrological systems and is recommended for the transfer of scientific understanding to ungauged or poorly gauged basins where calibration is not normally possible." -Pomeroy et al. (2007)

In this vein, we leverage a substantial dataset of diverse field observations. Particularly, we verify the snow model with data from manual plot- and basin-scale SWE surveys that are geographically distant from the local meteorological stations used to force the model. Such validation provides a fairer model skill assessment than the use of SNOTEL SWE observations, which are typically co-located with meteorological stations where errors due to meteorological interpolation should (theoretically) be lowest.

CHANGES:   With regard to calibration, we have added the following on Lines 137-138:

"The physically based model system was uncalibrated. Model decisions and parameters were chosen based on their successful application in previous studies."

With regard to model verification and meteorological forcing stations, we have added the following on Lines 538-541 (changes in bold):

"Notwithstanding, there are inherent strengths and weaknesses of the different validation data sets. **For example, automated SWE stations were often co-located with meteorological stations used to force the model; thus, the full potential for model error may not be evaluated at these locations. A fairer model assessment is possible when using data from the plot- and basin-scale snow surveys, which can be further from the local meteorological stations.**"

B)   We agree with the Reviewer that errors in spatial meteorological forcing fields can be significant. In fact, we conclude on Lines 548-550 (changes in **bold**) that "Overall, the model performed best in regions closest to precipitation gauges used to force the model (Fig. S1) and tended to slightly overestimate SWE at upper elevations (Table S3) **where no precipitation measurements are available**".

While resulting error in the baseline runs will "propagate through the climate projections", our characterization of problematic regions and times improves our ability to interpret our results. This is stated on Lines 529-530: "Improved model error characterization for the baseline (nominal) years is a critical step toward informed interpretation of the results of our climate change sensitivity analysis."

C)      The pre-processing library MeteoIO is an onboard component of the Alpine3D model system. The sensitivities of Alpine3D results to meteorological data coverage, and interpolation and model decisions are thoroughly addressed in Schlögl et al., (2016). The sensitivity of a physically based snow model to errors in forcing input is addressed in Raleigh et al. (2015). Additional assessment is beyond the scope of our study.

CHANGES:    The following sentences have been added on Lines 188-189: "The sensitivity of Alpine3D results to meteorological interpolation and model decisions are addressed in *Schlögl et al.,* [2016]."

References:

Pomeroy, J. W., Gray, D. M., Brown, T., Hedstrom, N. R., Quinton, W. L., Granger, R. J., & Carey, S. K. (2007). The cold regions hydrological model: a platform for basing process representation and model structure on physical evidence. *Hydrological processes*, *21*(19), 2650-2667.

Schlögl, S., Marty, C., Bavay, M., & Lehning, M. (2016). Sensitivity of Alpine3D modeled snow cover to modifications in DEM resolution, station coverage and meteorological input quantities. *Environmental Modelling & Software*, *83*, 387-396.

Raleigh, M. S., Lundquist, J. D., & Clark, M. P. (2015). Exploring the impact of forcing error characteristics on physically based snow simulations within a global sensitivity analysis framework. *Hydrology and Earth System Sciences*, *19*(7), 3153-3179.

Other specific comments:

Line 1: The abstract is very well written, with the exception of the first sentence. I did not understand your point until I read the sentence three times. Please consider restructuring it so that there is little to no ambiguity.

We have re-worded the first sentence of the abstract (changes in **bold**):

"In a warmer climate, the fraction of annual meltwater produced at high melt rates **in mountainous areas is** projected to decline due to a contraction of the **snow-cover season, causing melt to occur earlier and under lower energy conditions**."

Lines 90 – 93: I like your description of why you chose only to change changes in temperature.

Thank you.

Line 143: You describe that the model has an arbitrary number of layers. I found myself wanting more information about how the number of layers is/was decided. By the modeler or the model?

The number of snow layers is determined by the model. These model decisions are based on meteorological conditions during snowfall and simulated metamorphic processes (similar neighboring layers may be combined) and melt processes (layers may be removed as they melt). This is a now a common functionality in detailed snow models. We have decided to remove this sentence since 1) the model is described in detail in the Bartelt and Lehning (2002) citation, and 2) the number of snow layers is not directly relevant to our study or results.

Lines 242-243: How were the automated SWE measurements evaluated? This should be provided in the methods.

This is now clarified (changes in **bold**): Modeled SWE fields were evaluated against these station observations **using the RMSE and bias metrics described above**.

Lines 281 – 282: This is a very informative way to organize the elevation bands!

Thank you.

Lines 327 – 329: Please provide some of the result values for Snow Disappearance Date.

We do not quantitatively evaluate snow disappearance date in this study. We believe the (simulated) snow disappearance dates, averaged over elevation bands, are best visualized in Figure 5.

We use the metric to draw relative comparisons between years; however, an arithmetic average will have little physical meaning. We have expanded our description of inter-annual differences in snow disappearance date on Lines 322-326 (changes in **bold**):

"The AMJ period in the moderately wet year was > 2°C colder than the other years (Table 2) due to a series of large snowfall events in mid-April (Fig. 3) that prolonged snow-cover **well into June** (see Fig**s**. 2 **and 3**). **By comparison, snow-cover measured at the automated SWE stations generally disappeared in May in both the drier and average years (Fig. 3)."**

Lines 341 – 346: There are a lot of generalities (i.e. in general is used two times in three sentences). It would be informative to have to have detailed values and let the reader decide in general or not. In its present form, the reader simply has to assume what the authors are telling them with minimal supporting data/evidence.

These terms have been removed. We provide quantitative RMSE and bias metrics for the automated SWE stations and snow survey data. We used general terms to describe simulated snow depth relative to the envelope of measurements from six sensors each within four different model grid cells. We note that:

"Compared to the range of snow depth measured by six sensors at each of four sites in the forested Wolverton basin, the model accurately captured the seasonal snow depth dynamics, including maximum accumulation, the rate of depletion, and the date of snow disappearance (Fig. 4; note that simulated snow depth is generally within the measurement envelope)."

As shown in Fig. 4, there are very few exceptions to this statement, making it generally true.

Lines 359 – 361: I did not see the step-wise jump that you all are referring to here. I fully assume it is my error, but it might be helpful to annotate the jumps for your readers.

We now clarify how we interpret step-wise jumps from figure (changes in **bold**):

"Generally, maximum SWE occurred later with increasing elevation but progressed in a step-wise manner, often with little change over hundreds of vertical meters interspersed with abrupt jumps of one to two months (Fig. 5**; note the occasional large horizontal spacing between 'x' markers of adjacent elevation bands**)"

Lines 368 – 375: This is great data, and well organized.

Thank you.

Lines 445 – 455: This is an important policy/infrastructure component of this work. Consider adding an even more bold statement here. Something along the lines of: "The shift of SWE towards higher unmonitored elevations highlights the need for expanding the existing monitoring network to better manage water resources. For example, in the extremely low snow year of 2015, 1 March 47% of snow monitoring sites in the Willamette River basin, Oregon registered zero SWE, while snow was still present at higher elevations. (Sproles et al., 2017)".

Use the citation or not, but I recommend a stronger policy statement regarding the absence of SWE monitoring at higher elevations, and how the addition of these sites would be critical to monitor snowmelt response to warmer temperatures and for water resource management. Your efforts demonstrate why additional monitoring is important.

Thank you for the reference suggestion. We have added it:

"The results confirm previous findings in the U.S. Pacific Northwest that the current observing network design may be insufficient in a warmer world [Gleason et al., 2017; **Sproles et al., 2017**]."

Since our simulations are limited to three years and three stations, we believe this paragraph adequately communicates our findings, with relevant references to more comprehensive studies, and that further emphasis is not warranted.

Lines 481 – 483: This statement is an important discussion point that could be made stronger by adding a more recent citation and examples from the winters of 2014 and 2015.

We have added reference to a paper by Barnett and Pierce (2009):

Barnett, T. P., and D. W. Pierce (2009), Sustainable water deliveries from the Colorado River in a changing climate, Proceedings of the National Academy of Sciences, 106(18), 7334-7338.

We are not aware of any studies that have explicitly evaluated dam management decisions, particularly, winter water releases and impacts on summer storage, for the two years that the Reviewer suggests.

Lines 519 – 520: Throughout the paper I was expecting snowmelt lysimeters to be included, as either data or as discussion topic. I found the absence of any discussion of lysimeters in a paper focused on snowmelt to be incomplete. Are there no lysimeters in the study area? If not would this also be an infrastructure recommendation? At the minimum, I would suggest a brief explanation of how lysimeters could or could not improve the scientific understanding of snowmelt processes.

This is an excellent point. To our knowledge, there are no snowmelt lysimeters in Sequoia National Park. The data signal from a snowmelt lysimeter would inherently include both snowmelt runoff and rainfall that infiltrates and exits the snowpack. The daily snowmelt signal could be isolated if co-located with a precipitation gauge to extract rainfall response from the record. Measured daily SWE depletion from snow pillows is arguably a more direct estimate of snowmelt rate.

It is unclear how well snowmelt lysimeters work on steep slopes representative of mountainous watersheds. Also, lysimeters may work well at lower to middle elevations in maritime climates where the soil remains unfrozen, but problems have been reported in colder regions / higher elevations where sub-freezing soil-surface temperatures can cause ice to obstruct buried tipping buckets.

We have added the following sentence to Lines 514-516: "While not available in this region, snowmelt lysimeters may be useful additions to long-term research sites to better characterize variability and trends in the flux of water to the soil system."

Lines 529 – 531: This is a pretty critical point, especially since one of your primary findings is that the higher elevation snowpack will be more important under warmer conditions. The fact that the model performs better near forcing stations further supports my concerns modeled temperature and precipitation data not being calibrated/validated. This is an important component of modeling distributed snowpack. Doing so could perhaps improve model performance at higher elevations.

Thanks. Please see our associated response to the Reviewer's primary comment.

We now tie our model uncertainty at higher elevations to the Reviewer's previous point about a need to guide monitoring network design in a warmer climate (changes in **bold**):

Lines 548-553: "Overall, the model performed best in regions closest to precipitation gauges used to force the model (Fig. S1) and tended to slightly overestimate SWE at upper elevations (Table S3) **where no precipitation measurements are available**. **The results complement our**

**finding that the current precipitation and snowpack observation network may be insufficient in a warmer world where the majority of snow water resources shifts to higher, unmonitored elevations where snow model error is greatest.**"

Figure 2: Your color ramp for snow is counterintuitive. Red usually represents warning/drier conditions, but here it represents more snow. Additionally, if you are displaying sequential data it should be on a single color ramp.

This may be an issue of personal preference. In our color scheme, brighter / warmer colors are associated with higher values. An example of NASA using such a color scheme to map precipitation across the United States is provided below (https://www.nasa.gov/feature/goddard/2016/nasa-maps-el-ninos-shift-on-us-precipitation). My feeling is that as long as the color bar index is clear, and colors are easily distinguishable, then the color scheme used to visualize the data is effective.

[Figure]

Light, Adam, and Patrick J. Bartlein. "The end of the rainbow? Color schemes for improved data graphics." Eos 85.40 (2004): 385-391.
Also, http://colorbrewer2.org/, has color schemes that work really well. I believe there is a Matlab function for Colorbrewer as well.
https://www.mathworks.com/matlabcentral/fileexchange/34087-cbrewer---colorbrewerschemes-for-matlab

Figure 5: Again, colors are counterintuitive and should be redesigned.

Please see our response to the previous comment.

Figures 6, 7, and 10 are really informative. They provide a lot of information!

Thank you for the detailed and supportive review.

---

## Author Comment (AC2) · 16 Oct 2017

B. Henn (Referee)
bhenn@ucsd.edu

General Comments

This study examines the effect of projected climate warming on snow accumulation and melt rates along an elevation gradient and between wet and dry years. The authors use detailed snow and meteorology observations from Sequoia National Park to validate and drive physically based snow model simulations of historic and projected snowpack under warming. The simulations show that historic conditions are reproduced for three years of data with acceptable agreement to observations. They then show sensitivity of snow volumes and melt rates with elevation and wet/dry years, finding that snow volumes decline with warming at about 10%/degC overall with greater losses in the mid elevation coniferous forests, and that melt rates decline in areas of snow loss as melt now occurs earlier in the year. They also argue that extreme high melt rates increase under warming based on one of the three simulated years in which mid-winter melt events occurred more extensively.

The study reinforces previous work that shifts from snow to rain under warming will result in lower snowpacks in mid-elevation areas, and that the lower, more ephemeral snowpacks will melt more slowly and earlier in the winter. The elevation gradient and three years of varying precipitation are helpful in visualizing those effects across those important dimensions of the mountain hydroclimate, making the study a useful contribution.

The authors' claim that extreme melt rates (defined as high-quantile rates under the historic scenario) may increase under warming is novel and has flood risk implications. This claim may need to be phrased more carefully as the majority of the results presented show melt rates decreasing under almost all warming scenarios, and it also seems possible that the increases in melt rates reported are sensitive to the assumptions of the warming perturbations in this particular study. The authors should perhaps more clearly state some of these caveats.

The paper is well written and clearly organized and has high-quality figures.

Thank you for the thoughtful and supportive review. We address your comments and suggestions below.

Specific Comments

L254-262: The section on how longwave radiation was calculated under the perturbation scenarios is clear in terms of how it was done, but would benefit from more conceptual explanation about why this method is appropriate. For example, this method assumes that both RH and emissivity of the atmosphere do not change under the warming scenarios. While there is conceptual evidence to support (fairly) similar RH in a warmer climate, emissivity has a

dependence on temperature (Flerchinger, G. N., W. Xaio, D. Marks, T. J. Sauer, and Q. Yu (2009), Water Resour. Res., doi:10.1029/2008WR007394.)

Given that the authors cite midwinter melt rates as a key finding later in the study, and longwave has been implicated in driving midwinter melt (Lundquist, J. D., Dickerson-Lange, S. E., Lutz, J. a., & Cristea, N. C. (2013). Water Resources Research, http://doi.org/10.1002/wrcr.20504), justifying the perturbation assumptions around longwave and turbulent energy fluxes might benefit the paper.

We agree that this assumption should be stated and implications discussed. It is important to note that the air temperature influence on LW is by far the most dominant component of the Stefan-Boltzmann equation (effective air temperature is to the power of 4).

We have added the following sentence to the section where we first introduce the longwave radiation perturbation. Lines 265-266: "The *in-situ* atmospheric emissivity is assumed to be constant for the perturbed temperature scenarios."

We also discuss the implications of our assumption in the Discussion section where we discuss sources of uncertainty, and conclude that our results are somewhat conservative (Lines 557-560):

"Furthermore, by not perturbing the measured atmospheric emissivity used in the warmer scenarios, we may underestimate the longwave contribution to snowmelt. Atmospheric emissivity varies as a function of column-integrated temperature, specific humidity, and cloud structure above a site [Flerchinger et al., 2009]."

Thank you for the suggestion of the Flerchinger et al., (2009) citation.

L295-297: The explanation of the quantile analysis of melt rates could perhaps be better. If I am understanding correctly, the 99th percentile (for example) melt rates (over the whole spatial domain and year?) are calculated for the nominal case, and then this is repeated for the warming cases and the melt rates are compared?

We now better describe this analysis with the following sentences on Lines 299 – 303:

"For this analysis, the model domain was divided into three elevation bands: 1500 to 2250 m asl, 2250 to 2800 m asl, and >2800 m asl, and percentiles of daily snowmelt were computed for all grid cells in each elevation band. The analysis was conducted separately for each of the three water years and seven scenarios."

L376-389: Somewhere in the paper, discussion about why drier years were more susceptible snowpack loss to warming than wetter years might be warranted. Given that precipitation is fixed, were snow accumulations reduced more in the dry year because those storms were warmer (rain-snow level closer to mean domain elevation) than in wet years? i.e., is this a general finding or something specific to the storms in those years?

To clarify, we don't characterize the dry year as being particularly sensitive to warming. Rather, the wetter year is more resilient to warming than the other two years (an average and a moderately drier year). We explain this difference by the wetter year having substantially lower seasonal (AMJ) average air temperature; however, we do not explicitly analyze synoptic storm temperatures. While not shown, the series of spring storm events in 2010 (wetter year) were unseasonably cold. As simulated, these storms in spring 2010 brought substantial snowfall to low elevations even in the warmest scenario. Thus the phase change was resilient to simulated warming.

We provide a brief discussion of this on Lines 444-446: "The year with the most snowfall, characterized by late snowfall events and cold spring (AMJ) air temperatures, was slightly more resilient (-9.3% °C$^{-1}$) to warming than the drier or average snow years." These seasonal air temperature metrics are provided in Table 3.

L405-408: Perhaps I am misunderstanding Figure 9, but the statement that "Extreme melt rates (99th percentiles; downward-facing triangles in Fig. 9) actually increase (inferred from markers plotting above the 1:1 line) at elevations > 2800 m asl in all years (top panels) and in the drier year at all elevations (left panels)" doesn't quite seem to follow the data - most of the warming scenarios showed did not show increasing 99th percentile rates for the dry year at low elevations, nor did several of the scenarios for each year the high elevation zone. It is interesting that some extreme melt rates did increase, but focusing only on those scenarios that increased might overstate the robustness of this finding.

Thank you for pointing this out. This sentence has been reworded to clarify that we refer to "a majority of the simulations". In the drier year, the Reviewer is correct that only the top two elevation bands (>2250 m) had a majority of simulations in which the 99$^{th}$ percentile values were above the 1:1 line. Lines 412-415 (changes in **bold**):

"**For a majority of the simulations,** extreme melt rates (99$^{th}$ percentiles; downward-facing triangles in Fig. 9) actually increase (inferred from markers plotting above the 1:1 line) at elevations > 2800 m asl in all years (top panels) and in the drier year at elevations **>2250 m asl**."

This is now an accurate statement supported by the data.

L425: The discussion section could benefit from better organization and potentially using subsections to divide it among topics. The ordering of this section (discussing results, then flood and soil moisture implications, then caveats and other issues) seemed a bit meandering for me as a reader (implications for streamflow and soil moisture, processes which are not tested in this paper, seem like they should go last, for example).

We agree that the Discussion would benefit from reorganization. We have taken the Reviewer's suggestion to divide the Discussion Section into three sub-sections:

4.1. Snowmelt response to simulated warming

4.2.  Hydrologic Implications

**4.3. Sources of uncertainty and caveats**

L545: The mechanisms behind the reduction in snowpack with warming seem like they deserve greater discussion here. If I am understanding the experiment, there are only two mechanisms by which warming reduced meltwater volumes: precipitation falling as rain instead of snow, and increased sublimation. How important are these relative to each other in reducing meltwater volumes? Is increased sublimation significant at all or is it entirely the shift to rain? Some discussion of these mechanisms behind the results might be helpful to placing them in physical context.

We now clarify this point on Lines 576-580 (changes in **bold**):

"**The simulated reductions in snowmelt volume due to increased sublimation are very small compared to reductions caused by the warming induced shift from snow to rain. However,** by not considering blowing snow and subsequent sublimation losses (i.e., overestimating alpine snowpack), we may further underestimate snowpack sensitivity to warming."

Technical Corrections
Figure 1: Having repeated numbering of different types of stations is confusing - consider different ways of numbering sites.

Table 1 and Figure 1 have been updated such that each of the 29 stations / snow courses has a unique identifying number.